# Endogenous tagging using split mNeonGreen in human iPSCs for live imaging studies

**Mathieu C Husser[1], Nhat P Pham[1], Chris Law[1,2], Flavia RB Araujo[3], Vincent JJ Martin[1,3]\*, Alisa Piekny[1,2,3]\***

[1]Biology Department, Concordia University, Montreal, Canada; [2]Center for Microscopy and Cellular Imaging, Concordia University, Montreal, Canada; [3]Center for Applied Synthetic Biology, Concordia University, Montreal, Canada

**\*For correspondence:**
vincent.martin@concordia.ca (VM);
alisa.piekny@concordia.ca (AP)

**Competing interest:** The authors declare that no competing interests exist.

**Abstract** Endogenous tags have become invaluable tools to visualize and study native proteins in live cells. However, generating human cell lines carrying endogenous tags is difficult due to the low efficiency of homology-directed repair. Recently, an engineered split mNeonGreen protein was used to generate a large-scale endogenous tag library in HEK293 cells. Using split mNeonGreen for large-scale endogenous tagging in human iPSCs would open the door to studying protein function in healthy cells and across differentiated cell types. We engineered an iPS cell line to express the large fragment of the split mNeonGreen protein (mNG2$_{1-10}$) and showed that it enables fast and efficient endogenous tagging of proteins with the short fragment (mNG2$_{11}$). We also demonstrate that neural network-based image restoration enables live imaging studies of highly dynamic cellular processes such as cytokinesis in iPSCs. This work represents the first step towards a genome-wide endogenous tag library in human stem cells.

## eLife assessment

In this study, the authors develop a strategy for fluorophore-tagging endogenous proteins in human induced pluripotent stem cells (iPSCs) using a split mNeonGreen approach, and they conclude that the system will be appropriate for performing live imaging studies of highly dynamic cellular processes such as cytokinesis in iPSCs. Experimentally, the methods are **solid**, and the data presented support the authors' conclusions. Overall, these methodologies should be **useful** to a wide audience of cell biologists who want to study protein localization and dynamics at endogenous levels in iPSCs.

## Introduction

Since GFP was first described as a fluorescent reporter (*Chalfie et al., 1994*), the use of fluorescent proteins has become a standard method to study the localization and function of proteins inside living cells. Generally, reporter protein fusions are transiently or stably expressed from exogenous plasmids carrying their own promoter. However, expression levels can be quite high and variable from these strong, non-specific promoters (*Husser et al., 2022*). Further, over-expression can induce artifacts of localization and protein-protein interactions, making data challenging to interpret (*Gibson et al., 2013*; *Mahen et al., 2014*). With the advent of gene editing tools such as CRISPR/Cas9, the genes encoding fluorescent proteins can now be inserted directly into the genome at a desired locus (*Bukhari and Müller, 2019*; *Husser et al., 2021*). CRISPR/Cas9 is used to introduce a double-stranded break at the target site, which can be repaired by homology-directed repair (HDR) using a repair template

**eLife digest** The human body contains around 20,000 different proteins that perform a myriad of essential roles. To understand how these proteins work in healthy individuals and during disease, we need to know their precise locations inside cells and how these locations may change in different situations.

Genetic tools known as fluorescent proteins are often used as tags to study the location of specific proteins of interest within cells. When exposed to light, the fluorescent proteins emit specific colours of light that can be observed using microscopes. In a fluorescent protein system known as split mNeonGreen, researchers insert the DNA encoding two fragments of a fluorescent protein (one large, one small) separately into cells. The large fragment can be found throughout the cell, while the small fragment is attached to specific host proteins. When the two fragments meet, they assemble into the full mNeonGreen protein and can fluoresce.

Researchers can use split mNeonGreen and other similar systems to generate large libraries of cells where the small fragment of a fluorescent protein is attached to thousands of different host proteins. However, so far these libraries are restricted to a handful of different types of cells.

To address this challenge, Husser et al. inserted the DNA encoding the large fragment of mNeonGreen into human cells known as induced pluripotent stem cells, which are able to give rise to any other type of human cell. This then enabled the team to quickly and efficiently generate a library of stem cells that express the small fragment of mNeonGreen attached to different host proteins. Further experiments studied the locations of host proteins in the stem cells just before they divided into two cells. This suggested that there are differences between how induced pluripotent stem cells and other types of cells divide.

In the future, the cells and the method developed by Husser et al. may be used by other researchers to create atlases showing where human proteins are located in many other types of cells.

carrying the fluorescent marker (*Verma et al., 2017*). With this approach, the protein of interest is still expressed from its endogenous promoter, fused with the fluorescent protein. This enables the study of proteins at endogenous expression levels and provides more reliable measurements of protein behavior (*Dambournet et al., 2018*; *Doyon et al., 2011*; *Husser et al., 2022*; *Mahen et al., 2014*). Endogenous tagging is commonly done in model organisms such as *Saccharomyces cerevisiae*, for which whole-genome libraries of endogenous tags have been generated (*Huh et al., 2003*). However, gene editing in human cells is less widely used due to limitations in efficiency caused by transfection and HDR, among other issues. Because of these bottlenecks, the majority of human proteins have not been tagged and studied at the endogenous level. Although several efforts have been made to tag multiple proteins endogenously in various human cell lines (e.g. *Husser et al., 2022*; *Roberts et al., 2017*), the generation of a genome-wide library of tagged proteins in human cells requires higher throughput.

The recently developed self-complementing split fluorescent proteins can be used as tools for large-scale endogenous tagging (*Feng et al., 2017*; *Feng et al., 2019*; *Kamiyama et al., 2016*; *Tamura et al., 2021*; *Zhou et al., 2020*). In the split mNeonGreen system, the mNeonGreen protein is expressed as two separate fragments: a large fragment composed of the first ten beta-strands of mNeonGreen ($mNG2_{1-10}$) and a short fragment corresponding to the eleventh beta strand ($mNG2_{11}$; *Feng et al., 2017*). The two fragments have been engineered to form a functional fluorescent protein when co-expressed (*Feng et al., 2017*). This interaction is characterized by a high complementation efficiency and is irreversible once the reconstituted protein has folded (*Feng et al., 2019*; *Köker et al., 2018*). Moreover, the split mNeonGreen2 protein retains most of the brightness of the original mNeonGreen (*Feng et al., 2017*). This system allows for easy and efficient endogenous tagging with $mNG2_{11}$ in cells where $mNG2_{1-10}$ is constitutively expressed (*Cho et al., 2022*; *Leonetti et al., 2016*; *Mahdessian et al., 2021*). The $mNG2_{11}$ fragment is only 16 amino acids long, so it can be inserted into the genome by HDR using a repair template with short homology arms (generally 40–80 bp), which can be purchased commercially as a single-stranded oligo-deoxynucleotide (ssODN). This enables large-scale tagging using commercially synthesized ssODNs and sgRNAs to target multiple proteins in parallel (*Cho et al., 2022*; *Leonetti et al., 2016*). However, this approach requires the generation of a

parental cell line that constitutively expresses the mNG2$_{1-10}$ fragment. This has been done by random lentiviral integration followed by antibiotic selection, which is fast but generates a heterogeneous population where the expression of the large fragment is inconsistent across the cell population and is subject to epigenetic silencing (*Cabrera et al., 2022*). Large-scale endogenous tagging in HEK293 cells was recently achieved by the OpenCell project, with 1310 proteins tagged to date (*Cho et al., 2022*). However, while this library will be a valuable resource for the community, there is a need to study protein function in other cell types, where the mechanisms regulating biological processes could vary drastically.

Since their discovery in 2007, human induced pluripotent stem cells (iPSCs) have become a popular tool to study human cells in developmental contexts and to identify disease-causing mutations, amongst other valuable applications (*Shi et al., 2017*; *Takahashi et al., 2007*). iPSCs are derived from somatic cells that are reprogrammed to be self-renewing and pluripotent. These cells can be cultured and differentiated into any desired cell type in vitro using specific protocols (e.g. *Grancharova et al., 2021*; *Hong and Do, 2019*; *Oceguera-Yanez et al., 2022*). Despite this incredible resource, few studies have investigated cellular processes in human stem cells and differentiated cell types with high spatiotemporal resolution (*Dambournet et al., 2018*; *Viana et al., 2023*). For example, several studies describe the regulation of mitosis and cytokinesis in mouse embryos and embryonic stem cells (*Chaigne et al., 2020*; *Chaigne et al., 2021*; *Paim and FitzHarris, 2022*), but not in human stem cells. Since most of our knowledge of human cell cytokinesis is derived from diseased and/or transformed cell lines, the field will benefit greatly from studying iPSCs before and after differentiation in an isogenic context, particularly since they represent 'healthy' human cells (*Dambournet et al., 2018*; *Drubin and Hyman, 2017*).

Here, we used the split mNeonGreen system for endogenous tagging in human iPSCs (*Figure 1A*). First, we engineered a human iPS cell line that constitutively expresses the mNG2$_{1-10}$ fragment. We validated this cell line extensively and named it 'smNG2-P' (split mNeonGreen2 parental cell line). As a proof-of-concept, we efficiently targeted multiple genes for endogenous tagging with mNG2$_{11}$, and clonally isolated several tagged iPS cell lines. To facilitate this process, we developed protocols for efficient single-cell isolation by FACS (fluorescence-activated cell sorting) and screening of edited clones by Nanopore sequencing. Finally, we show how the endogenously tagged iPS cell lines can be used for live imaging studies of cytokinesis, which is a highly dynamic cellular process. Timelapse imaging of several weakly expressed cytokinesis genes that were endogenously tagged in iPSCs revealed new insights into how cytokinesis occurs in these cells. Imaging the more weakly expressed genes required the use of an image restoration algorithm to alleviate cell toxicity and obtain high-quality images with high temporal resolution. This work provides the foundation for a genome-wide endogenous tag library in human stem cells and provides protocols to efficiently generate and study endogenous tags in human iPSCs.

## Results

### Generation of a split mNeonGreen iPS cell line for efficient endogenous tagging

Our first goal was to generate a human iPS cell line where mNG2$_{1-10}$ is constitutively expressed. To do this, we generated a repair template that contains a mNG2$_{1-10}$ expression cassette for integration into the AAVS1 locus, based on a previously published design (*Figure 1B*; *Oceguera-Yanez et al., 2016*). This cassette contains the CAG promoter to drive high levels of mNG2$_{1-10}$ expression, and resist silencing at the AAVS1 locus over time and during differentiation (*Luo et al., 2014*; *Oceguera-Yanez et al., 2016*). The cassette also includes the gene for Puromycin resistance, which is expressed upon integration at the AAVS1 locus via a splicing acceptor and self-cleaving T2A peptide (*Figure 1B*). We introduced this mNG2$_{1-10}$ expression cassette into the 201B7 cell line, which was generated by retroviral transduction of reprogramming factors into dermal fibroblasts and is well-characterized (*Takahashi et al., 2007*). After transfecting 201B7 human iPSCs with the AAVS1-mNG2$_{1-10}$ repair template and an AAVS1-targeting Cas9/sgRNA RNP (ribonucleoprotein) complex, the cells were selected for Puromycin resistance and single-cell clones were isolated by FACS for screening. Clones were first screened by qPCR for the presence of the mNG2$_{1-10}$ gene and the absence of the Ampicillin resistance (AmpR) gene, which was part of the backbone of the repair template (*Figure 1—figure supplement*

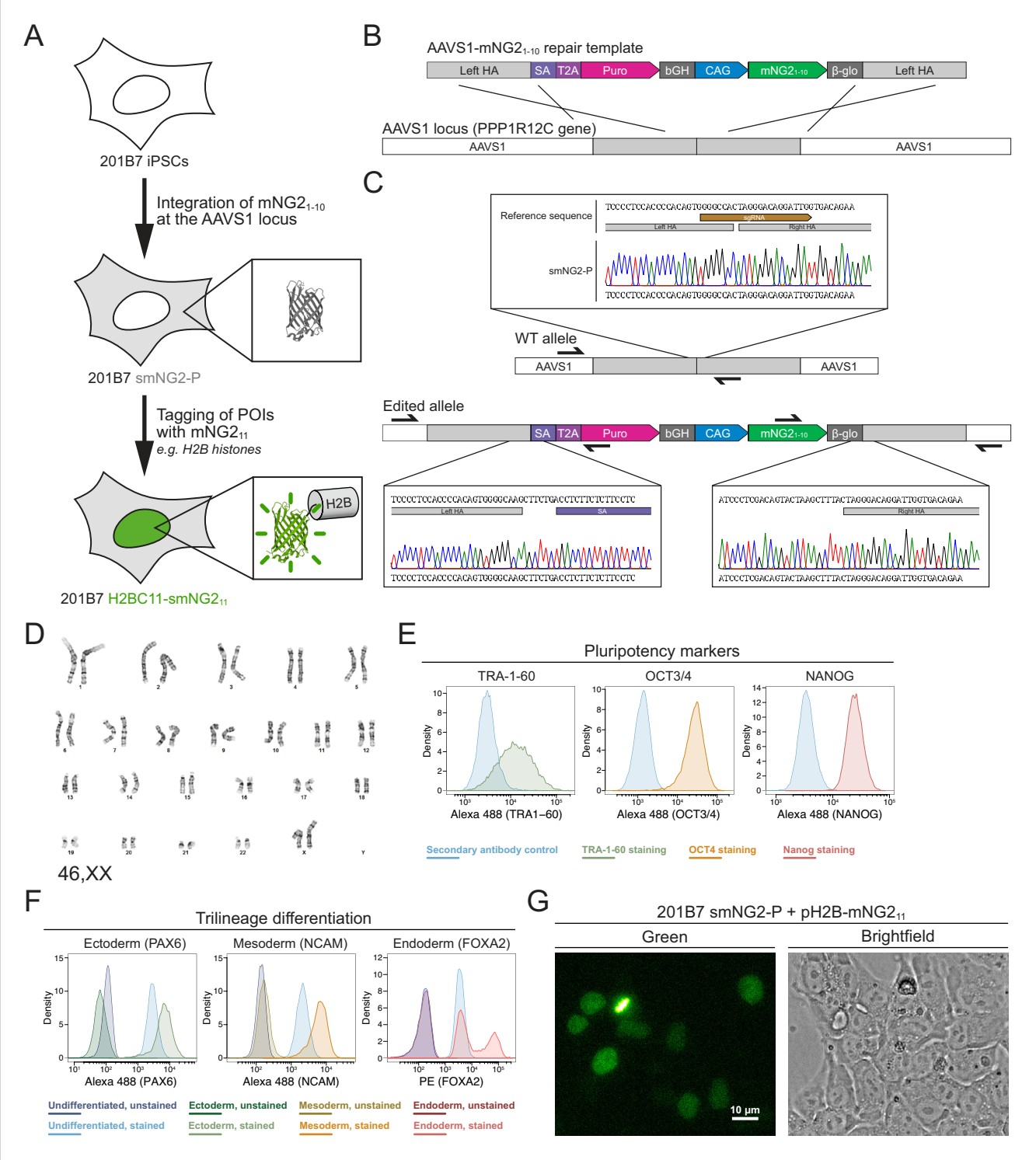

**Figure 1.** Generation of a split mNeonGreen human induced pluripotent stem (iPS) cell line. (**A**) A schematic representation of the split tagging strategy used in this study. An expression cassette carrying the mNG2₁₋₁₀ fragment was integrated at the AAVS1 locus in 201B7 iPSCs to generate the parental split mNeonGreen2 cell line (smNG2-P). Proteins of interest were tagged endogenously with the mNG2₁₁ fragment to visualize their expression and localization. The structure of mNeonGreen was generated from PDB (Protein Data Bank, structure identifier 5 LTP; *Clavel et al., 2016*). (**B**) A schematic of the mNG2₁₋₁₀ expression cassette is shown with the CAG promoter (blue) and sequences for stable expression (dark gray), Puromycin resistance marker (pink), and homology arms (light gray) for integration at the AAVS1 locus. Abbreviations: SA = splicing acceptor, T2A=Thosea asigna virus 2A peptide, bGH = bovine growth hormone poly-adenylation signal, β-glo=rabbit β-globin poly-adenylation signal. (**C**) Sequencing chromatograms show

*Figure 1 continued on next page*

*Figure 1 continued*

the edited AAVS1 allele junctions in the smNG2-P cell line (bottom) compared to the WT allele (top). (**D**) Representative G-banding karyotype of the smNG2-P cell line shows that the edited cells have a normal karyotype (46, XX). (**E**) Flow cytometry plots show smNG2-P cells stained for pluripotency markers TRA-1–60 (left, green), OCT3/4 (center, yellow), and NANOG (right, red) or a secondary antibody control (blue). (**F**) Flow cytometry plots show differentiated smNG2-P cells stained for PAX6 (ectoderm; left, green), NCAM (mesoderm; center, yellow), and FOXA2 (endoderm; right, red) compared to undifferentiated cells (blue) and unstained controls (dark colors). (**G**) Fluorescent and brightfield images show fluorescence complementation after transfecting split mNeonGreen2 parental cell line (smNG2-P) cells with a plasmid expressing H2B fused to $mNG2_{11}$. The scale bar is 10 microns.

The online version of this article includes the following source data and figure supplement(s) for figure 1:

**Figure supplement 1.** Screening of $AAVS1-mNG2_{1-10}$ clones.

**Figure supplement 1—source data 1.** Source data for *Figure 1—figure supplement 1A—D*.

**Figure supplement 1—source data 2.** Folder containing the raw and annotated gel images used for *Figure 1—figure supplement 1F*, G, and H.

**Figure supplement 2.** Validation of the split mNeonGreen2 parental cell line (smNG2-P) cell line.

**Figure supplement 2—source data 1.** Source data for *Figure 1—figure supplement 2A*.

**Figure supplement 2—source data 2.** Folder containing the raw and annotated gel images used for *Figure 1—figure supplement 2O*.

*1A–D*). Clones that were positive for $mNG2_{1-10}$ and negative for the AmpR gene were then screened for integration at the AAVS1 locus by junction PCR (*Figure 1—figure supplement 1E–H*). We found 6 clones that were heterozygous for $AAVS1-mNG2_{1-10}$ integration, and one clone was selected for full validation (clone 28).

## Validation of the smNG2-P iPS cell line

We further validated clone 28 to ensure that the cells are still healthy and carry only the desired $AAVS1-mNG2_{1-10}$ integration. Sequencing of the AAVS1 locus revealed that one allele contains the $mNG2_{1-10}$ expression cassette, while the other allele is unedited (*Figure 1C*). To ensure that only one copy of the cassette was integrated properly, we measured the genomic copy number for $mNG2_{1-10}$ and for the AmpR gene by digital PCR (dPCR). Our data confirmed the presence of a single copy of $mNG2_{1-10}$ with no ectopic integration (*Figure 1—figure supplement 2A*). Next, 12 sites predicted to be susceptible to off-target editing based on previous studies and prediction software were selected for sequencing (*Concordet and Haeussler, 2018*; *Hsu et al., 2013*; *Wang et al., 2014*). We found no differences in the sequence of the 12 off-target sites between the WT 201B7 cell line and our edited cells (*Figure 1—figure supplement 2B–M*). To ensure that the clone 28 cells have a normal genome, we performed G-banding karyotyping and found no chromosomal abnormalities (46, XX karyotype; *Figure 1D*). To verify that clone 28 cells are undifferentiated, we used immunofluorescence staining and flow cytometry with antibodies specific for TRA-1–60, OCT3/4, and NANOG (*Figure 1E*). We found that all three pluripotency markers were expressed. We also found that clone 28 cells retain pluripotency by successfully differentiating them into ectoderm, mesoderm, and endoderm, as determined by immunofluorescence staining and flow cytometry with antibodies specific for PAX6 (ectoderm), NCAM (mesoderm), and FOXA2 (endoderm; *Figure 1F*). Clone 28 cells also have normal iPSC morphology and tested negative for mycoplasma contamination (*Figure 1—figure supplement 2N–O*). Finally, we verified that mNeonGreen fluorescence could be reconstituted in the presence of the $mNG_{11}$ fragment. Transfection of clone 28 cells with a plasmid expressing H2B (histone) fused to the $mNG_{11}$ fragment resulted in the expression of a fluorescent signal localized to chromatin as expected for the functional complementation of the $mNG2_{1-10}$ and $mNG2_{11}$ fragments (*Figure 1G*). After these validation and quality control steps, we chose clone 28 as the $AAVS1-mNG2_{1-10}$ cell line, which we hereafter refer to as 'smNG2-P' (split mNeonGreen2 parental cell line).

## Efficient endogenous tagging with $mNG2_{11}$ in smNG2-P cells

Since fluorescence could be reconstituted by expressing the $mNG_{11}$ fragment in the smNG2-P cell line, we aimed to integrate $mNG2_{11}$ into different endogenous loci. We selected 17 genes for tagging, some of which had been previously endogenously tagged with the split mNG system (e.g. ACTB, TUBA1B, KRT18; *Cho et al., 2022*) or with full-length fluorescent proteins (e.g. H2BC11, ACTB, ANLN, RHOA; *Husser et al., 2022*; *Roberts et al., 2017*), while other genes had not been tagged previously (e.g. RACGAP1, KRT5, TUBB3, FOXA2). We also expected some of these genes to be expressed at varying levels in iPSCs. The expression levels for these genes are shown on a graph of

RNA-seq data from *Iwasaki et al., 2022*; *Figure 2—figure supplement 1A*. While ACTB and TUBA1B should be highly expressed, there is a shift to lower expression levels for KRT18, RHOA, and H2BC11, while SOX2, CDH1, NES, TUBB3, ANLN, and RACGAP1 are all weakly expressed. Finally, NCAM1, FOXA2, PAX6, TBXT, KRT5, and KRT14 should be silent in iPSCs as they are only expressed in other cell types (*Figure 2—figure supplement 1A*). Altogether, these genes cover a range of expression levels, localization patterns, and involvement in different cellular processes. For each protein, we selected the N- or C-terminus for tagging based on published studies, and we used a short flexible linker (GGG; *Feng et al., 2017*; *Kamiyama et al., 2016*; *Leonetti et al., 2016*) to minimize disruption of the tagged protein. We designed ssODNs to introduce the mNG2$_{11}$ tag at these loci, and transfected them into smNG2-P cells along with gene-specific Cas9/sgRNA RNP complexes. Eight days after transfection, we quantified the proportion of WT, indel, and tagged alleles in the edited cell populations by Nanopore sequencing (*Figure 2A–B*). We found that Cas9 targeting was efficient for all loci, as shown by the high proportion of indels in the edited cell populations. However, the proportion of tagged alleles was variable, with 37.5% of CDH1 alleles tagged, while only 0.69% of TUBA1B alleles were tagged (*Figure 2B*). This data shows that while endogenous tagging with mNG2$_{11}$ can be highly efficient, it varies considerably for each locus.

Next, we monitored the reconstitution of mNeonGreen fluorescence in edited populations by flow cytometry and fluorescence microscopy. As expected for genes that are expressed in iPSCs, we observed a fluorescent signal in H2BC11, ACTB, TUBA1B, ANLN, RHOA, KRT18, SOX2, NES, and TUBB3-tagged populations (*Figure 2—figure supplement 1B–T*). Tagged cell populations were enriched by FACS, and microscopy was used to determine if their localization was consistent with prior studies (*Figure 2C*). As expected, the fluorescent signal from H2B histone was nuclear (*Viana et al., 2023*), β-actin formed filaments and was enriched at the cortex (*Roberts et al., 2017*; *Viana et al., 2023*), α-tubulin was cytosolic and enriched in filaments and mitotic spindles (*Roberts et al., 2017*; *Viana et al., 2023*), anillin was nuclear and enriched in the furrow of dividing cells (*Hesse et al., 2012*; *Husser et al., 2022*), RhoA was cytosolic and weakly enriched at the cortex (*Husser et al., 2022*; *Yonemura et al., 2004*), keratin 18 formed cortical filaments (*Maurer et al., 2008*), SOX2 was nuclear (Allencell.org; *Strebinger et al., 2019*), Nestin formed distinct filaments (*Kuang et al., 2019*), and β–3-tubulin was weakly expressed and cytosolic (*Guo et al., 2010*; *Turaç et al., 2013*). For these proteins, the mean fluorescence intensity of tagged cells correlated well with their expected relative expression levels (*Figure 2—figure supplement 1A* and U). Despite the high tagging efficiency observed for CDH1, we did not observe any fluorescent signal by flow cytometry (*Figure 2—figure supplement 1K*). However, microscopy revealed that E-cadherin-mNG2$_{11}$ was enriched at cell junctions (*Figure 2—figure supplement 1V*; Allencell.org; *Aban et al., 2021*; *Cumin et al., 2022*). This suggests that weakly expressed proteins may require different methods to verify expression following endogenous tagging. However, we did not observe any fluorescent signal from tagged Cyk4 (RACGAP1 gene) by flow cytometry or microscopy (*Figure 2—figure supplement 1G*). The lack of signal could be due to the low tagging efficiency (*Figure 2B*), and/or the weak expression of Cyk4 in stem cells (*Figure 2—figure supplement 1A*; *Iwasaki et al., 2022*). There was no fluorescent signal in the KRT5, KRT14, PAX6, TBXT, NCAM1, and FOXA2-edited populations, as expected for differentiation markers (*Figure 2B* and *Figure 2—figure supplement 1*). We observed nuclear fluorescence by microscopy in FOXA2-tagged cells after induction into endoderm (*Figure 2—figure supplement 1W–X*), but we did not investigate the expression of the others as this goes beyond the scope of our study.

## Efficient clonal isolation and screening of tagged iPS cell lines

Next, we isolated five clonal cell lines expressing H2B histones, β-actin, α-tubulin, anillin, and RhoA tagged with mNG2$_{11}$, as these proteins have distinct localization patterns during cell division (*Beaudet et al., 2017*; *Husser et al., 2022*; *Rodrigues et al., 2015*; *van Oostende Triplet et al., 2014*). We optimized a protocol to efficiently recover single-cell clones after FACS (shown in *Figure 3A*), resulting in an average recovery of 61.4% in 96-well plates across the five genes (*Figure 3—figure supplement 1A*). To facilitate the screening of clonal recovery in 96-well plates, we developed an ImageJ macro that automatically identifies positive and negative wells from 96-well plate images (*Figure 3—figure supplement 1B–G*). Isolated clones were then screened based on their genotype by multiplexed Nanopore sequencing. We found that clones had diverse genotypes with alleles

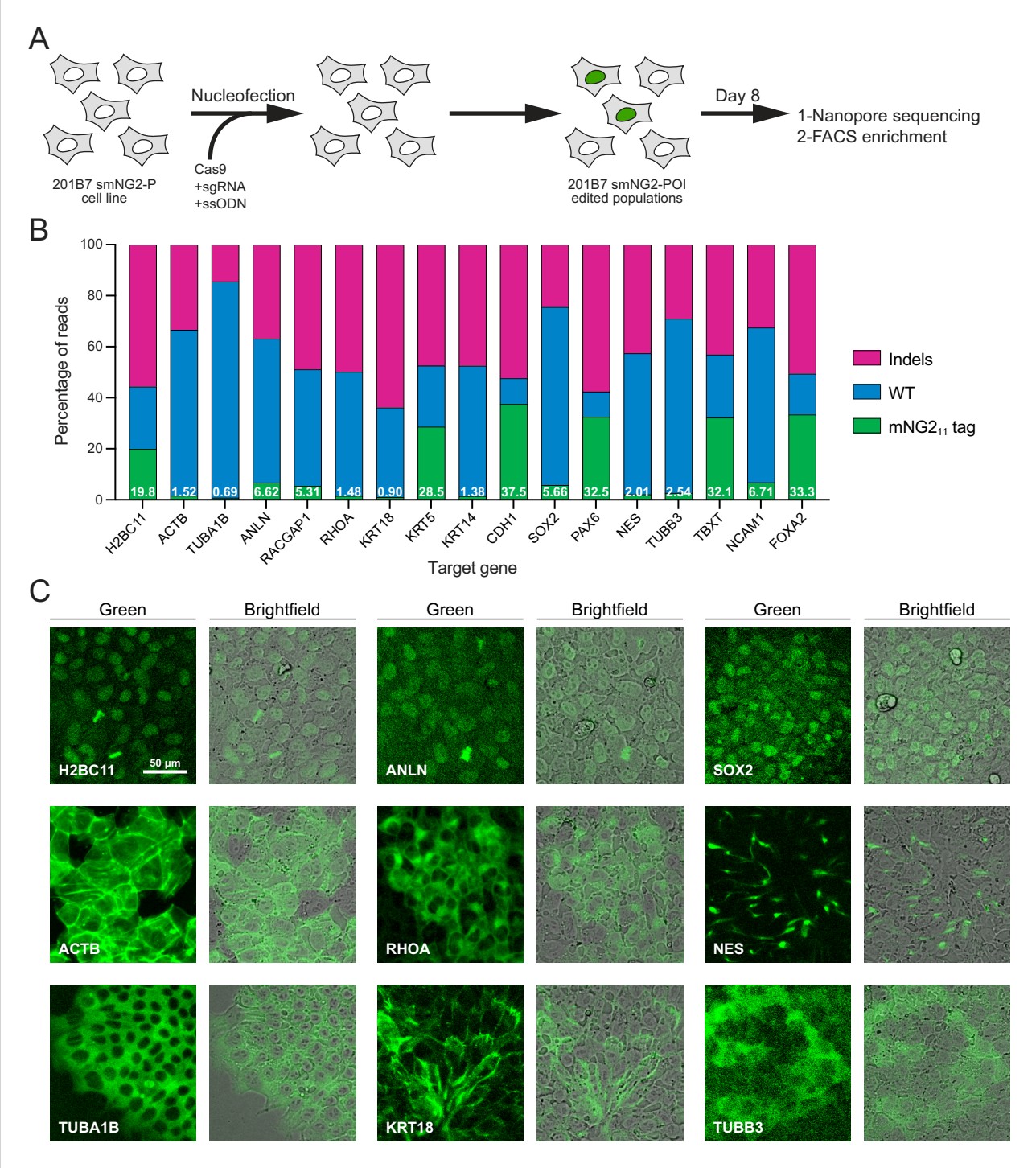

**Figure 2.** Efficient endogenous tagging with mNG2_{11} in split mNeonGreen2 parental cell line (smNG2-P) cells. (**A**) A schematic shows the workflow used to tag proteins of interest with the mNG2_{11} fragment. smNG2-P cells were transfected with Cas9/sgRNA RNPs and a single-stranded repair template (ssODN) to integrate mNG2_{11} at a target locus. After recovery, the edited populations were frozen and assessed by Nanopore sequencing and flow cytometry. (**B**) A bar graph shows the distribution of alleles (WT, indels, or mNG2_{11}-tagged) in edited populations. The percentage of mNG2_{11} alleles is indicated in white for each gene. (**C**) Fluorescent and brightfield images show populations of tagged cells after enrichment by FACS as indicated. The scale bar is 50 microns.

The online version of this article includes the following source data and figure supplement(s) for figure 2:

**Source data 1.** Source data for **Figure 2B**.

*Figure 2 continued on next page*

*Figure 2 continued*

**Figure supplement 1.** Fluorescent signal from endogenous mNG2$_{11}$ tags.

**Figure supplement 1—source data 1.** Source data for *Figure 2—figure supplement 1U*.

containing mNG2$_{11}$, mutated mNG2$_{11}$, indel mutations, and WT sequences (*Figure 3B*). The diversity of genotypes amongst tagged cells suggests that clonal isolation is important to obtain a high-quality homogeneous population of cells for further study. We obtained both homozygous and heterozygous clones carrying smNG2-anillin and smNG2-RhoA, and we found homozygous clones that were brighter than heterozygous ones (*Figure 3—figure supplement 1H–I*). This suggests that the complementation between mNG2$_{1-10}$ and mNG2$_{11}$ occurs efficiently over a range of mNG2$_{11}$ expression, since anillin is expressed weakly and RhoA is expressed more strongly in iPSCs. We also observed some homozygous clones that were not brighter than the corresponding heterozygous clones, which could be due to undetected by-products of CRISPR or clonal variation in protein expression. This variability underlines the importance of screening and validating edited clones. For each tagged protein, a single clone was selected for further study (*Figure 3C–G*). These cell lines were also tested for mycoplasma contamination (*Figure 3—figure supplement 1J*). Fluorescent images of the final mNG2$_{11}$-tagged cell lines acquired using confocal swept-field microscopy are shown in *Figure 3H*. Consistent with the expected localization for these proteins, H2B was nuclear during interphase and localized to condensed chromatin during mitosis, actin was cortical in both interphase and mitotic cells, tubulin was cytosolic during interphase and localized to mitotic spindles, anillin was nuclear during interphase and enriched in the cleavage furrow during cytokinesis, and RhoA was cytosolic during interphase and cortically enriched during mitosis.

## Image restoration for live imaging of cellular processes in iPSCs

The endogenously tagged iPS cell lines can be used to study the mechanisms controlling different cellular processes in healthy cells and how they vary with cell type. Our knowledge of human cytokinesis is derived from transformed and/or cancerous, differentiated cell lines and has not been studied in human pluripotent stem cells. Cytokinesis is dynamic and requires imaging over long periods of time (tens to hundreds of minutes) at frequent intervals. To study cytokinesis in human iPSCs, we imaged the mNG2$_{11}$-tagged cell lines from metaphase onwards. After a few minutes of imaging using standard optical settings, iPSCs stopped dividing and detached from the coverslip, independently of laser wavelength (data not shown), suggesting that iPSCs are more sensitive to photo-toxicity than transformed and cancerous cell lines. Moreover, endogenous protein levels are low, requiring higher exposure times. To overcome this issue, we optimized the settings to reduce exposure time and laser power, decrease z-stack depth, and increase imaging intervals so that cells could be imaged for more than 80 min. However, while we could detect sufficient levels of actin using these imaging conditions, proteins that are more weakly expressed were more challenging to detect (*Figure 4A*). Indeed, the signal-to-noise ratio was low for H2B, α-tubulin, anillin, and RhoA in images acquired with low-exposure settings that supported cell survival, compared to images acquired using high-exposure settings that did not support cell viability (*Figure 4A*). To overcome this issue, we used deep learning-based image restoration to obtain high-resolution images from those with low signal-to-noise ratios caused by low-exposure settings. We trained a CARE (Content-Aware image REstoration) neural network on sets of matched high- and low-exposure images for each cell line (*Figure 4B*; *Weigert et al., 2018*), and used the trained model to restore timelapse movies acquired with low-exposure settings (*Figure 4C–G*). The signal-to-noise ratio was drastically improved for timelapse images of H2B, α-tubulin, anillin, and RhoA (*Figure 4C–G*). Most importantly, this approach allowed us to image live iPSCs with high temporal resolution.

With the ability to image key cytoskeletal and cytokinesis regulators such as RhoA and anillin, we characterized cytokinesis in iPSCs for the first time. In metazoans, cytokinesis initiates by the assembly of a RhoA-dependent contractile ring in anaphase which ingresses during telophase to pinch in the membrane (*Figure 5A*), and then transitions to a stable midbody for abscission (*Ozugergin and Piekny, 2022*; *Pollard and O'Shaughnessy, 2019*). Anillin crosslinks key components of the ring, membrane and spindle for ring positioning, ingression, and midbody formation (*Green et al., 2012*; *Piekny and Maddox, 2010*). We recently found that ring assembly kinetics and ingression varies among a few transformed and/or cancerous human cell types, and correlates with the localization of

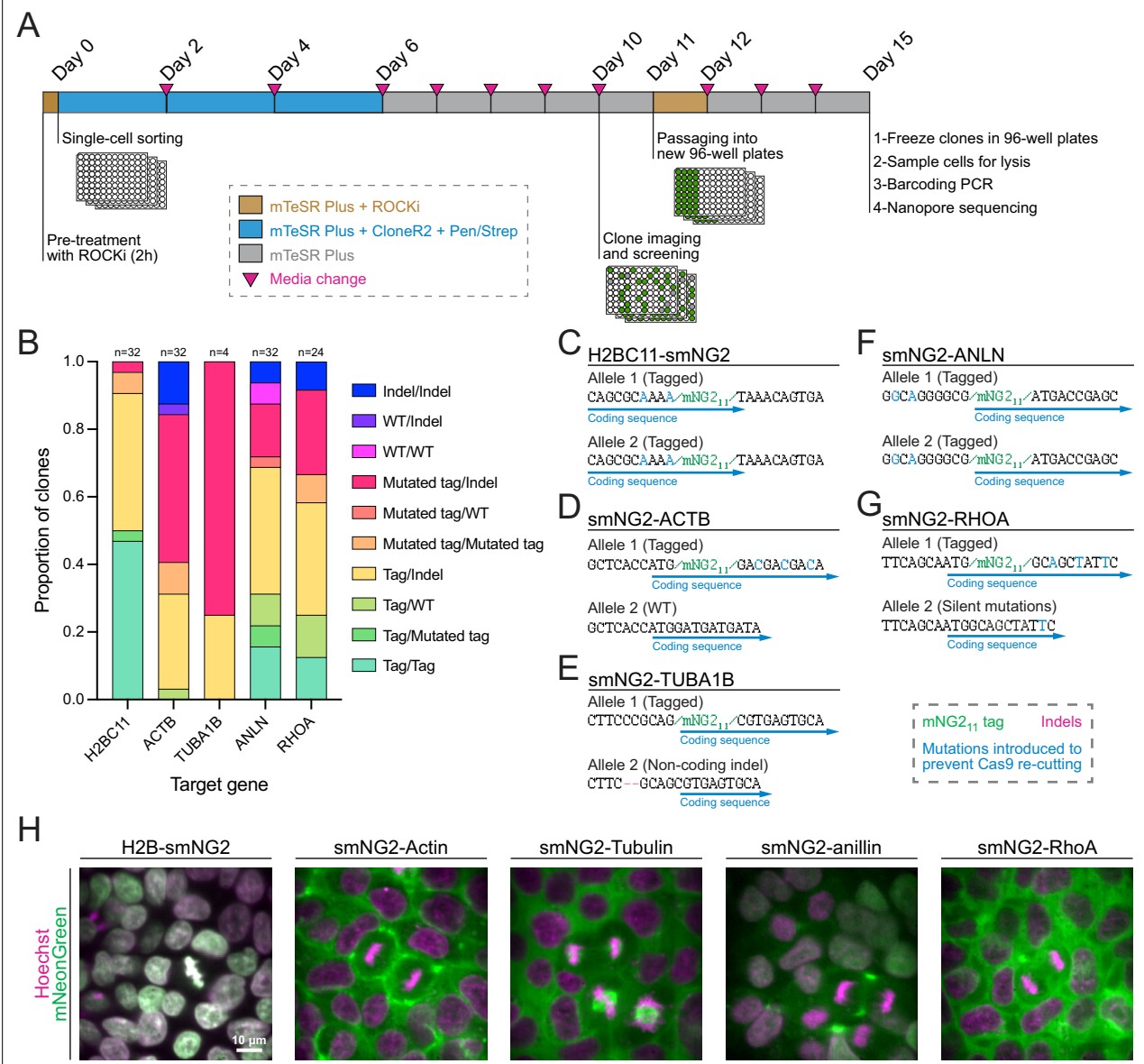

**Figure 3.** Efficient clonal isolation and screening of tagged induced pluripotent stem (iPS) cell lines. (**A**) The protocol used for single-cell isolation and recovery of tagged iPSC clones is shown. Tagged cell populations were pre-treated with ROCK inhibitor Y-27632 for 2 hr before sorting individual cells into 96-well plates by FACS. Cells were kept in recovery media for 6 days, then colonies were screened on day 10 to select those for further passaging into 96-well plates on day 11. Fully-grown clones were frozen on day 15, and cells were genotyped by barcoding PCR followed by Nanopore sequencing. (**B**) A stacked bar graph shows the distribution of genotypes inferred from multiplexed Nanopore sequencing for isolated clones of tagged H2BC11, ACTB, TUBA1B, ANLN, and RHOA (sample size is indicated above each bar). (**C–G**) The genotypes of the final tagged cell lines are shown: H2BC11-smNG2 (**C**), smNG2-ACTB (**D**), smNG2-TUBA1B (**E**), smNG2-ANLN (**F**), and smNG2-RHOA (**G**). The target gene coding sequence is in blue, indel mutations are in red and mNG$_{11}$ tag in green. (**H**) Fluorescent images show the localization of smNG2 (green) and DNA (stained with Hoechst; magenta) in the final tagged cell lines. The scale bar is 10 microns.

The online version of this article includes the following source data and figure supplement(s) for figure 3:

**Source data 1.** Source data for *Figure 3B*.

**Figure supplement 1.** Clonal isolation and screening of tagged clones.

**Figure supplement 1—source data 1.** Source data for *Figure 3—figure supplement 1A, H, I*.

**Figure supplement 1—source data 2.** Folder containing the raw and annotated gel images used for *Figure 3—figure supplement 1J*.

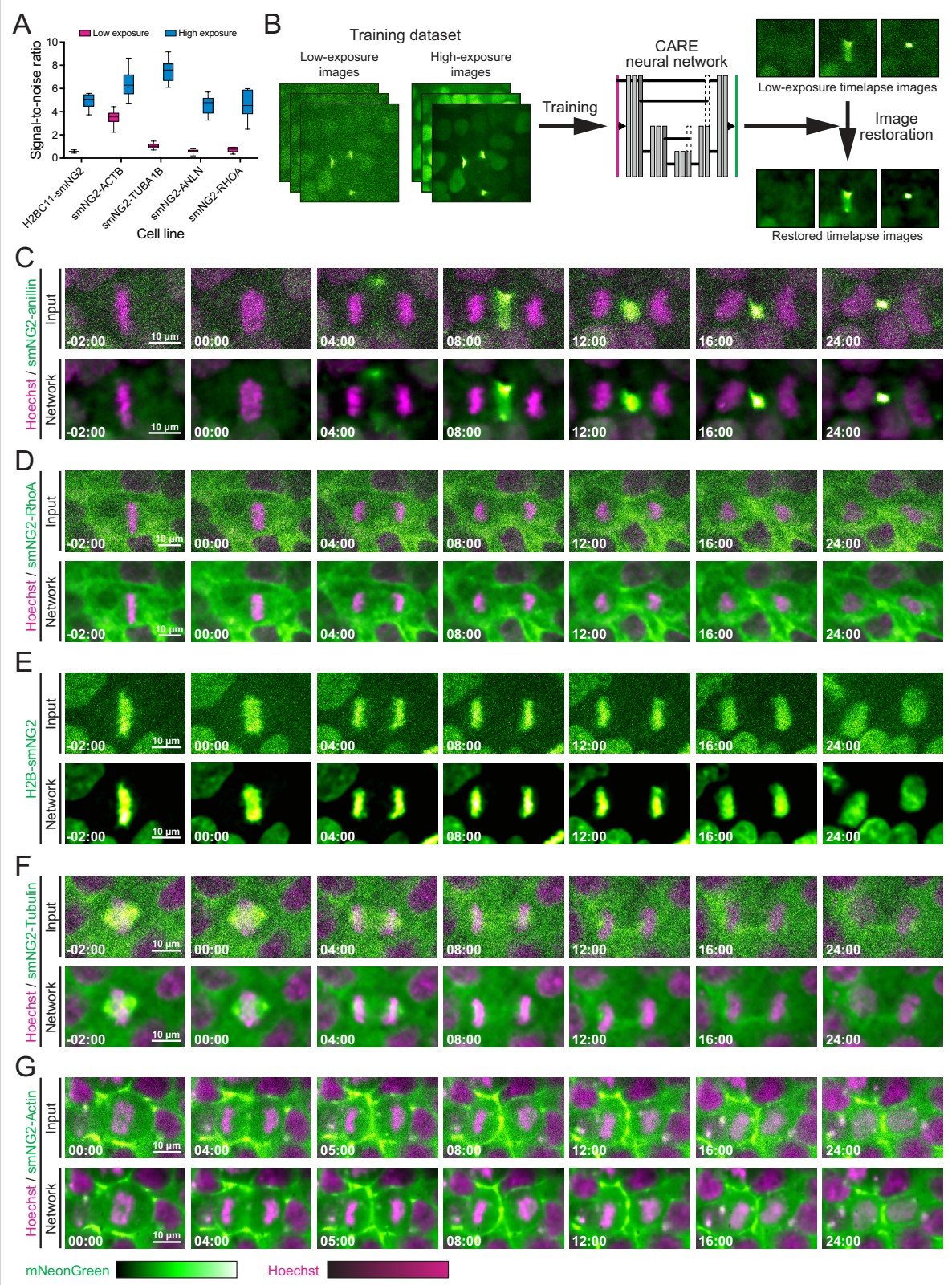

**Figure 4.** Image restoration for live imaging and quantitative measurements in induced pluripotent stem cells ( iPSCs). (**A**) A graph shows the signal-to-noise ratio measured from low- (pink) and high-exposure (blue) images of the tagged iPS cell lines as indicated (n≥7 for each condition). (**B**) A schematic shows the training and application of the Content-Aware image REstoration (CARE) neural network for image restoration. The neural network developed by *Weigert et al., 2018* was trained on sets of low- and high-exposure images for each tagged cell line and used to restore timelapse

*Figure 4 continued on next page*

*Figure 4 continued*

movies. (**C–G**) Comparisons of raw (Input; top panel) and restored timelapse images (Network; bottom panel) by the CARE neural network are shown for iPSCs expressing smNG2-anillin (**C**), smNG2-RhoA (**D**), H2B-smNG2 (**E**), smNG2-Tubulin (**F**), and smNG2-actin (**G**) undergoing cytokinesis (smNG2 in green; DNA stained with Hoechst in magenta). The scale bars are 10 microns, and time is relative to anaphase onset (00:00). The box plot in A shows the median line, quartile box edges and the minimum and maximum value whiskers.

The online version of this article includes the following video and source data for figure 4:

**Source data 1.** Source data for *Figure 4A*.

**Figure 4—video 1.** Comparison of raw (left) and restored (right) timelapse images of a smNG2-ANLN induced pluripotent stem (iPS) cell undergoing cytokinesis.

https://elifesciences.org/articles/92819/figures#fig4video1

**Figure 4—video 2.** Comparison of raw (left) and restored (right) timelapse images of a smNG2-RHOA induced pluripotent stem (iPS) cell undergoing cytokinesis.

https://elifesciences.org/articles/92819/figures#fig4video2

**Figure 4—video 3.** Comparison of raw (left) and restored (right) timelapse images of a H2BC11-smNG2 induced pluripotent stem (iPS) cell undergoing cytokinesis.

https://elifesciences.org/articles/92819/figures#fig4video3

**Figure 4—video 4.** Comparison of raw (left) and restored (right) timelapse images of a smNG2-TUBA1B induced pluripotent stem (iPS) cell undergoing cytokinesis.

https://elifesciences.org/articles/92819/figures#fig4video4

**Figure 4—video 5.** Comparison of raw (left) and restored (right) timelapse images of a smNG2-ACTB induced pluripotent stem (iPS) cell undergoing cytokinesis.

https://elifesciences.org/articles/92819/figures#fig4video5

**Figure 4—video 6.** Comparison of raw (left) and restored (right) timelapse images of a smNG2-ACTB induced pluripotent stem (iPS) cell showing colony-scale re-arrangements of the actin network after cytokinesis.

https://elifesciences.org/articles/92819/figures#fig4video6

anillin (*Husser et al., 2022*). Characterizing cytokinesis in iPSCs will be an important starting point to build new knowledge of how cytokinesis is regulated in healthy, non-transformed cells. First we measured the duration of contractile ring ingression in anillin-tagged iPSCs (18.6±4.0 min; *Figure 5B*). We then measured the localization of anillin, RhoA, and actin in metaphase and throughout cytokinesis. In metaphase cells, we found that anillin is cytosolic, while RhoA is weakly enriched at the cortex and actin is strongly enriched at the cortex (*Figures 4C, D, G , and 5C*). During mitotic exit, anillin accumulates at the equatorial cortex ~4 min after anaphase onset, while RhoA accumulates by ~6 min, and actin remains strongly cortical with some equatorial enrichment by ~4–5 min (*Figures 4C, D, G , and 5D*). We also found that actin is enriched at cell junctions, which remodel between adjacent cells during cytokinesis (*Figure 5—figure supplement 1A*). In an extreme example, junctions disappeared, and foci appeared adjacent to the site of ingression, likely resulting from the mechanical response to the forces generated by the furrow (*Figure 5—figure supplement 1A*). We then measured the breadth of cortically enriched anillin and RhoA at the onset of ingression and found that anillin forms a narrow peak (4.1±0.7 μm), while RhoA localizes more broadly (6.6±2.2 μm; *Figure 5E–F*). The same result was obtained when measuring furrow breadth as a percentage of cortex length, showing that the difference between RhoA and anillin localization is independent of cell size (*Figure 5—figure supplement 1B*). Since RhoA and anillin are expected to closely co-localize (*Piekny and Maddox, 2010*), we investigated this discrepancy by looking at the central spindle and astral microtubules in high-exposure images of tagged tubulin cells (*Figure 5G*). We found that cells have a poorly-organized central spindle in anaphase (*Figure 5G–H*), and the astral microtubules appear to contact the equatorial cortex (*Figure 5G*). This data reveals new differences in how cytokinesis occurs in iPSCs and demonstrates the utility of endogenous tags to study cellular processes by live imaging, which also can be done in differentiated cell types with isogenic backgrounds.

## Discussion

Recent advances in genetic engineering have accelerated the development of tools for fundamental research. Endogenous tags are particularly valuable as they enable the visualization of protein behavior

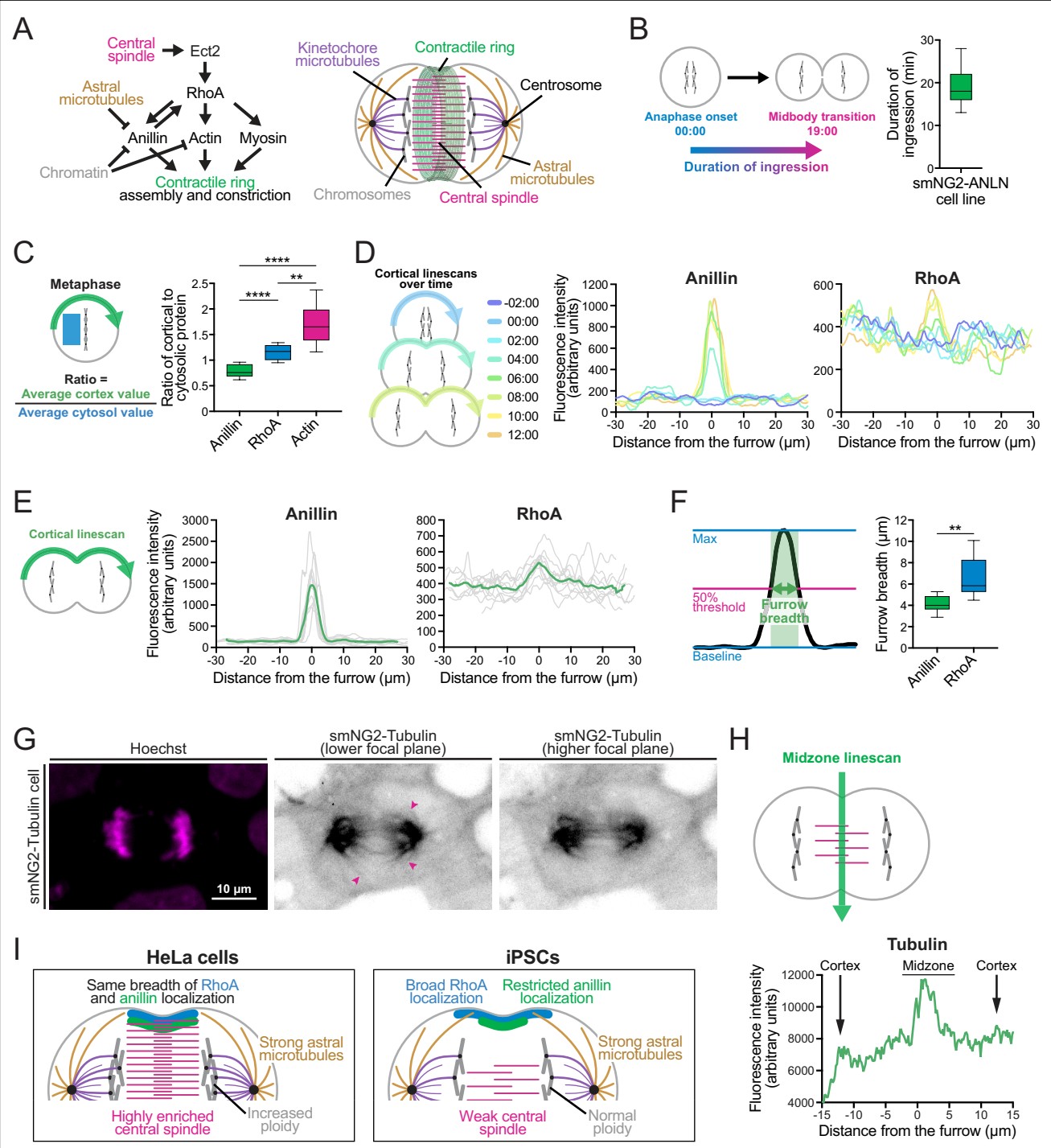

**Figure 5.** Cytokinesis is uniquely regulated in induced pluripotent stem cells (iPSCs). (**A**) Schematics show the core pathways regulating contractile ring assembly and ingression (left) and their localization inside a late anaphase cell (right). (**B**) A schematic (left) shows how the duration of ingression was measured in anillin-tagged iPSCs (right; n=39). (**C**) A schematic (left) shows how the ratio of cortical to cytosolic protein was measured in metaphase cells for smNG2-anillin, smNG2-RhoA, and smNG2-actin (right; n=10 for each). (**D**) A schematic (left) shows the location and timing of the linescans used to plot the fluorescence intensity along the cortex in single iPSCs for smNG2-anillin (center) and smNG2-RhoA (right). Timepoints are shown in different colors as indicated in the scale. (**E**) A schematic (left) shows the location of the linescans used to plot the intensity of fluorescence along the cortex at the onset of furrowing for smNG2-anillin (center) and smNG2-RhoA (right; n=10 for each). (**F**) A schematic (left) shows how the breadth of protein localization at the equatorial cortex at furrow initiation was calculated for smNG2-anillin and smNG2-RhoA (n=10 for each). (**G**) Fluorescent images show the location of DNA (left; stained with Hoechst) and smNG2-Tubulin (gray scale) in a cell. For tubulin, a lower focal plane (center) is shown to highlight astral microtubules (indicated by pink arrows), and a higher focal plane (right) is shown to highlight the central spindle. The scale bar is 10 microns. (**H**)

*Figure 5 continued on next page*

*Figure 5 continued*

A schematic (top) shows the location of the linescan used to plot the fluorescence intensity along the midzone of a cell expressing smNG2-tubulin in anaphase (bottom). (**I**) A schematic shows a comparison of the conventional view of cytokinesis (HeLa cell; left) and an iPS cell (iPSCs; right). HeLa cells have a strongly enriched central spindle, which keeps the equatorial of active RhoA narrow, and astral microtubules that extend all the way to the cortex and restrict the localization of anillin to the same zone as RhoA. iPSCs have a weakly enriched central spindle, resulting in a broader zone of active RhoA, and prominent astral microtubules that keep anillin in a narrow equatorial zone. Box plots in B, C, and F show the median line, quartile box edges and the minimum and maximum value whiskers. Statistical significance was determined by Brown-Forsythe and Welch's ANOVA follow by Dunnett's T3 test for C and Welch's t-test for F (ns, not significant; *p≤0.05; **p≤0.01; ***p≤0.001; ****p≤0.0001).

The online version of this article includes the following source data and figure supplement(s) for figure 5:

**Source data 1.** Source data for *Figure 5B, C, D, E, F and H*.

**Figure supplement 1.** Protein localization in induced pluripotent stem cells (iPSCs) during cytokinesis.

**Figure supplement 1—source data 1.** Source data for *Figure 5—figure supplement 1B*.

in live cells at endogenous expression levels. Self-complementing split fluorescent proteins have been used for efficient endogenous tagging, and this approach was recently used for the construction of the first library of endogenous tags in HEK293 cells, with 1310 proteins tagged in mixed populations enriched by FACS (*Cho et al., 2022*). While this library is a powerful resource for the community, researchers will need to generate single clones of individual tagged proteins before they can be studied. Further, the knowledge generated by this library is restricted to this single-cell type. Given the predicted diversity in mechanisms controlling cellular processes across cell types, there is a need to also generate endogenous tags in stem cells that can be differentiated into multiple cell types. In this study, we generated a fully validated human iPS cell line expressing mNG2$_{1-10}$ for efficient endogenous tagging with mNG2$_{11}$ in human stem cells capable of differentiating into any cell type. The 201B7 smNG2-P cell line is heterozygous for the mNG2$_{1-10}$ expression cassette which was integrated at the AAVS1 safe-harbor locus. Compared to lentiviral delivery, this design minimizes the risk of mNG2$_{1-10}$ silencing and provides a high and consistent expression of mNG2$_{1-10}$ across all cells in the population, even over time and during differentiation (*Oceguera-Yanez et al., 2016*). We tagged multiple genes with mNG2$_{11}$, with efficiencies ranging from 0.69% to 37.5% measured by Nanopore sequencing (*Figure 2B*). Tagging efficiency with mNG2$_{11}$ in iPSCs was lower than in mouse embryos (40 to 100% of injected embryos; *O'Hagan et al., 2021*), and lower than tagging with GFP$_{11}$ in HEK293 cells (<1 to 56% tagged alleles; *Cho et al., 2022*). However, endogenous tagging with mNG2$_{11}$ was overall more efficient than endogenous tagging with full-length mEGFP in iPSCs (mostly <0.1 to 4%, and up to 24% GFP-positive cells; *Roberts et al., 2017*). Although it is difficult to compare editing efficiencies across studies, endogenous tagging by HDR is very inefficient. Our results are consistent with iPSCs being more challenging to edit compared to other cell types, and mNG2$_{11}$ integrating at higher frequency than larger tags. The split mNeonGreen system also enables endogenous tagging on the scale of hundreds to thousands of proteins in parallel. For comparison, endogenous tagging with full-length fluorescent proteins requires the assembly of large repair templates, which limits the number of proteins that can be tagged at once. We also found that low endogenous expression levels can limit the detection of tagged proteins by flow cytometry, similar to previous studies (*Cho et al., 2022*; *Leonetti et al., 2016*; *O'Hagan et al., 2021*), requiring microscopy with highly sensitive cameras or detectors. Alternative methods such as fixation and antibody staining can also be used for more sensitive detection of weakly expressed endogenous tags (*O'Hagan et al., 2021*). Importantly, the expression levels of 9 mNG2$_{11}$-tagged proteins in iPSCs (*Figure 2C* and *Figure 2—figure supplement 1U*) were consistent with RNA-seq data (*Figure 2—figure supplement 1A*; *Iwasaki et al., 2022*). Since these proteins showed unique localizations patterns consistent with their function, it is unlikely that the mNG2$_{11}$ tag affected their function. Interestingly, Nestin and β–3-tubulin were expressed at low levels in undifferentiated iPSCs, despite being commonly used as ectodermal and neuronal differentiation markers. This result is consistent with a previous report of their expression in some iPS cell lines (*Kuang et al., 2019*) and with published RNA-seq and proteomic data for the 201B7 cell line (*Iwasaki et al., 2022*). The detection of a broader range of endogenous protein expression in live cells will require further improvements in the brightness of fluorescent proteins or in the sensitivity of detectors.

We also found that edited cells had diverse genotypes caused by gene editing, including alleles carrying mutated mNG2$_{11}$ tags and indel mutations, warranting clonal isolation. Such mutations have

been reported previously but are not fully understood (*Burgio and Teboul, 2020*; *Skryabin et al., 2020*). Single-nucleotide mutations may come from mutations introduced during ssODN synthesis. Meanwhile, ssODN re-arrangements could be caused by microhomologies, repair by NHEJ instead of HDR, or a combination of NHEJ and HDR repair (*Skryabin et al., 2020*). Unwanted editing outcomes are ignored when looking at pooled populations, and we recommend isolating and screening clonal cell lines to perform quantitative studies of desired cellular processes, as they provide a more reliable readout of protein expression, where variations in signal between cells within a population represent true cell-to-cell variability. Clonal isolation is notoriously inefficient in iPSCs, which are programmed to undergo dissociation-induced apoptosis upon loss of attachment or cell-cell contact (*Bhargava et al., 2022*; *Chen and Pruett-Miller, 2018*; *Singh, 2019*; *Tristan et al., 2023*; *Watanabe et al., 2007*). We optimized a protocol for efficient single-cell recovery of 201B7 iPSCs after FACS, with up to 80% of wells in a 96-well plate showing clonal growth after 10 days (*Figure 3A* and *Figure 3— figure supplement 1A*). We also created a macro for image-based colony screening in 96-well plates (*Figure 3—figure supplement 1B–G*), and used multiplexed Nanopore sequencing for high-throughput screening of selected colonies, as many amplicons can be sequenced simultaneously (*Whitford et al., 2022*). Automated instrumentation could be used to further increase the scale of clonal cell isolation, and to isolate cell lines where transcriptionally silent genes are tagged endogenously (*Roberts et al., 2019*). After clonal isolation, we found that the cells in each edited line had consistent, stable fluorescence. Further, we observed different levels of fluorescence across a range of protein expression levels (*Figure 2—figure supplement 1A–T*), showing that mNG2$_{1-10}$/mNG2$_{11}$ complementation is comparable to full-length fluorescent proteins. Importantly, the stability of fluorescence levels in endogenously tagged cell lines and the absence of localization artifacts compared to over-expressed transgenes makes this system attractive for quantitative measurements in live cells (*Husser et al., 2022*).

The goal of endogenous tagging in iPSCs is to enable live imaging studies of diverse cellular processes for comparative studies among different human cell types. We found that iPSCs are particularly sensitive to phototoxicity, making it difficult to image weakly expressed endogenous tags over longer periods of time or with short time intervals. This sensitivity has not been previously reported in human or in mouse stem cells, likely because imaging conditions are rarely reported and vary with different setups. In addition, timelapse imaging often involves the use of over-expressing transgenes or dyes, which generate higher fluorescent signal intensity and enable the use of optical settings with lower laser power and exposure time (*Chaigne et al., 2021*; *Hesse et al., 2012*; *Roberts et al., 2017*). Since most proteins are expressed weakly in the endogenous context, improvements in fluorophore brightness, imaging conditions, or image processing are required. Since many labs have access to spinning disk confocal or epifluorescence widefield microscopes, image restoration methods provide a way to obtain high-quality images of iPSCs without compromising on temporal resolution. We used CARE, a content-aware image restoration neural network developed by *Weigert et al., 2018*, where training datasets are created with matched images obtained using low- and high-exposures (*Figure 4*). With this network, cells can be imaged over extended periods of time with high temporal resolution using low exposure settings, and image files can be restored to generate high-quality movies (*Figure 4* and *Figure 4—videos 1–6*).

As a proof-of-concept, we characterized cytokinesis in live human iPSCs for the first time. Most of our knowledge of human cell cytokinesis was obtained from studies using HeLa cells, yet the mechanisms regulating this process are expected to vary with cell type (*Husser et al., 2022*; *Ozugergin and Piekny, 2022*). As a result, we do not have detailed knowledge of how cytokinesis occurs in most healthy human cell types. CARE allowed us to measure the timing of cytokinesis and the localization of core cell components (actin, tubulin) and cytokinesis proteins (RhoA, anillin) in live human iPSCs (*Figure 5*). We found that anillin-tagged iPSCs completed ingression in 18.6±4.0 min; faster than HepG2 and HEK293 cells, slower than HCT116 cells, and similar to HeLa cells (*Figure 5B*; *Husser et al., 2022*). Anillin is recruited to a narrow band at the equatorial cortex ~4 min after anaphase onset, while RhoA is also enriched but only after ~6 min and is more broadly distributed in iPSCs, while in HeLa cells they are both enriched after ~6 min and have the same breadth (*Figure 5D–E*; *Husser et al., 2022*; *Piekny and Maddox, 2010*). Active RhoA is generated at the equatorial cortex by the guanine nucleotide exchange factor Ect2 which is activated by Cyk4 at the central spindle (*Koh et al., 2022*; *Mahlandt et al., 2021*; *Yüce et al., 2005*). In HeLa cells, the central spindle starts to form

immediately after anaphase onset and reaches the cortex by late anaphase (~6 min) (*van Oostende Triplet et al., 2014*). Thus, the weak central spindle in anaphase iPSCs may explain the broad localization of RhoA (*Figure 5G–H*). In support of this, perturbations that weaken the central spindle in HeLa cells result in more diffuse Ect2-Cyk4 complexes and a broader zone of active RhoA (*Adriaans et al., 2019*; *Kotýnková et al., 2016*; *Su et al., 2011*; *Yüce et al., 2005*). Active RhoA is also required for the cortical recruitment of anillin, yet the localization of anillin in iPSCs is more narrow compared to the other human cell lines that we previously characterized (*Figure 5F*; *Husser et al., 2022*; *Piekny and Glotzer, 2008*). In iPSCs, the astral microtubules extended to the cortex in the equatorial region, which could restrict anillin localization. In HeLa cells, we previously showed that perturbations that cause an increase in astral microtubules decreases the breadth of anillin (*van Oostende Triplet et al., 2014*). However, interactions with other proteins or phospholipids could further restrict the localization of anillin in iPSCs (*Ozugergin and Piekny, 2022*). Previously, we speculated that the breadth of anillin localization inversely correlates with the speed of ingression. The iPSCs seem to differ from this model, suggesting that other factors should also be considered, such as the distribution of other actin crosslinkers or cortical flows (*Khaliullin et al., 2018*; *Leite et al., 2020*; *Osório et al., 2019*; *Reymann et al., 2016*; *Sobral et al., 2021*; *Spira et al., 2017*). Thus our findings reveal how cytokinesis occurs in a non-transformed human stem cell, and provide a framework to reveal how the mechanisms controlling cytokinesis vary with different cell types in an isogenic context.

Our work combines approaches to alleviate some of the challenges of gene editing in human stem cells. High transfection and editing efficiencies can be achieved in iPSCs by using electroporation and delivering Cas9/sgRNA RNP complexes instead of plasmids (*Kim et al., 2014*; *Liang et al., 2015*). Using a short tag that can be carried on single-stranded repair templates (ssODNs) bypasses the requirement for cloning dsODNs with large homology arms. Tagging efficiency is further increased by using NHEJ inhibitors (*Chu et al., 2015*; *Maruyama et al., 2015*; *Maurissen and Woltjen, 2020*; *Schimmel et al., 2023*). Cells with mutated $mNG2_{11}$ tags can be eliminated by screening single-cell clones based on genotype, using single-cell isolation and image-based colony screening in 96-well plates, combined with multiplexed genotyping of clones by Nanopore sequencing. Finally, we showed how iPSCs could be imaged with high temporal resolution using CARE for studies of cytokinesis, an essential dynamic cellular process. Altogether, this work provides the basis for a high-quality endogenous tag library in human iPSCs, which will be used to study protein function in human stem cells and across human cell types in vitro.

## Materials and methods

**Key resources table**

| Reagent type (species) or resource | Designation | Source or reference | Identifiers | Additional information |
|---|---|---|---|---|
| Strain, strain background (*Escherichia coli*) | DH5α | Genome Canada | | |
| Strain, strain background (*Escherichia coli*) | ER2925 | Genome Canada | | Methylase-deficient strain |
| Cell line (*Homo sapiens*) | 201B7 | ATCC | ACS-1023 | |
| Cell line (*Homo sapiens*) | smNG2-P | This paper | | |
| Antibody | Mouse monoclonal anti-NANOG | DSHB | PCRP-NANOGP1-2D8 | 1:25 dilution |
| Antibody | Mouse monoclonal anti-OCT3/4 | Santa Cruz Biotechnology | sc-5279 | 1:20 dilution |
| Antibody | Mouse monoclonal Alexa488-conjugated anti-TRA-1–60 | STEMCELL Technologies | 60064AD | 1:20 dilution |
| Antibody | Rabbit polyclonal Alexa488-conjugated anti-mouse | Thermo Fisher Scientific | A-11059 | 1:400 |
| Antibody | Mouse monoclonal anti-PAX6 | DSHB | PAX6 | 1:45 |
| Antibody | Mouse monoclonal anti-NCAM | DSHB | 5.1H11 | 1:240 |

*Continued on next page*

*Continued*

| Reagent type (species) or resource | Designation | Source or reference | Identifiers | Additional information |
|---|---|---|---|---|
| Antibody | Mouse monoclonal PE-conjugated anti-FOXA2 | BD Biosciences | 561589 | 1:50 |
| Recombinant DNA reagent | pNCS-mNeonGreen | Allele Biotechnology | ABP-FP-MNEONSB | |
| Recombinant DNA reagent | pAAVS1-P-CAG-GF | *Oceguera-Yanez et al., 2016* | Addgene #80491 | |
| Recombinant DNA reagent | pAAVS1-P-CAG-mNG2$_{1-10}$ | This paper | Addgene #206042 | |
| Recombinant DNA reagent | pH2B-mNG211 | This paper | Addgene #206043 | |
| Sequence-based reagent | sgRNA | Sigma-Aldrich | | Listed in *Supplementary file 3* |
| Sequence-based reagent | sgRNA | Synthego | | Listed in *Supplementary file 3* |
| Sequence-based reagent | ssODN | BioCorp | | Listed in *Supplementary file 4* |
| Sequence-based reagent | Primers | Thermo Fisher Scientific | | Listed in *Supplementary file 5* and 6 |
| Sequence-based reagent | Hydrolysis probes | IDT | | Sequences listed in the Materials and Methods |
| Peptide, recombinant protein | Cas9-eGFP | STEMCELL Technologies | 76006 | |
| Peptide, recombinant protein | Cas9 | Synthego | SpCas9 2NLS Nuclease | |
| Commercial assay or kit | P3 Nucleofection kit | Lonza | V4SP-3096 | |
| Commercial assay or kit | STEMdiff Trilineage Differentiation Kit | STEMCELL Technologies | 05230 | |
| Commercial assay or kit | QuickExtract | Biosearch Technologies | QE0905T | |
| Commercial assay or kit | Ligation Sequencing Kit V14 | Oxford Nanopore Technologies | SQK-LSK114 | |
| Commercial assay or kit | NEBNext Companion Module | New England Biolabs | E7180S | |
| Commercial assay or kit | RNaseP reference assay | Thermo Fisher Scientific | 4403328 | |
| Commercial assay or kit | QuantStudio 3D Digital PCR Master Mix V2 | Thermo Fisher Scientific | A26358 | |
| Commercial assay or kit | QuantStudio 3D Digital PCR 20 K Chip Kit V2 | Thermo Fisher Scientific | A26316 | |
| Chemical compound, drug | CloneR2 | STEMCELL Technologies | 100–0691 | Single-cell recovery reagent |
| Chemical compound, drug | NU7441 | Tocris Bioscience | 3712 | NHEJ inhibitor |
| Chemical compound, drug | SCR7 | Xcess Biosciences | M60082 | NHEJ inhibitor |
| Chemical compound, drug | Hoechst 33342 | Thermo Fisher Scientific | H1399 | |
| Software, algorithm | CARE | *Weigert et al., 2018* | https://github.com/CSBDeep/CSBDeep; *Schmidt et al., 2023* | Version 0.7.2 |
| Software, algorithm | Fiji | NIH | https://imagej.net/software/fiji/ | Version 2.3 |
| Software, algorithm | CRISPResso2 | *Clement et al., 2019* | https://github.com/pinellolab/CRISPResso2; *Pinello Lab, 2024* | Version 2.2.12 |
| Other | 35 mm polymer coverslip dish | Cellvis | D35-20-1.5P | Coverslips for live imaging |

## Cell culture

201B7 iPSCs were obtained from ATCC (catalog number ACS-1023). Cells were cultured in six-well plates coated with iMatrix-511 Silk (Iwai North America Inc) as per the manufacturer's protocol, in mTeSR Plus media (STEMCELL Technologies). The cells were kept at 37 °C and 5% $CO_2$. The cells were passaged every 3–4 days by dissociating them using Accutase (STEMCELL Technologies) according to the manufacturer's protocol. After dissociation, the cells were resuspended and 50,000–100,000 cells were transferred into fresh matrix-coated plates with mTeSR Plus media containing 10 µM Y-27632 (ROCK inhibitor, STEMCELL Technologies). Fresh media without ROCK inhibitor was fed the next day, and the media was replaced daily until the next passage.

## Constructs

All plasmids were maintained and cloned in *Escherichia coli* DH5α unless specified otherwise. Point mutations were introduced into pNCS-mNeonGreen (Allele Biotechnology) by PCR-based site-directed mutagenesis to convert it to mNeonGreen2 (*Feng et al., 2017*). The $mNG2_{1-10}$ fragment was then amplified by PCR. The left and right homology arms for the AAVS1 locus and the puromycin gene were amplified individually from the pAAVS1-P-CAG-GFP plasmid (Addgene #80491; *Oceguera-Yanez et al., 2016*). These fragments were cloned into the pYTK089 backbone by Golden Gate using a standard protocol (Addgene #65196; *Lee et al., 2015*). Since the CAG promoter was recalcitrant to amplification by PCR, it was digested directly from pAAVS1-P-CAG-GFP using BclI and EcoRI, gel extracted, and ligated into the pre-digested promoter-less plasmid to generate the pAAVS1-P-CAG-$mNG2_{1-10}$ repair template (Addgene #206042). For BclI digestion, the plasmids were extracted from methylase-deficient *E. coli* ER2925. pH2B-$mNG2_{11}$ (Addgene #206043) was generated by amplifying the H2B gene from an H2B-mRuby plasmid (*Beaudet et al., 2017*) with primers designed to add $mNG2_{11}$ on the end of H2B. This gene was then cloned into pEGFP-N1 digested with BamHI and NotI to replace EGFP with H2B-$mNG2_{11}$.

## Transfection

Cells were edited using transfection. Cells were treated with the NHEJ inhibitors NU7441 (2 µM; Tocris Bioscience) and SCR7 (1 µM; Xcess Biosciences) for 4 hr before and 48 hr after nucleofection, while 10 µM ROCK inhibitor was added to the media 2 hr before nucleofection to increase recovery. For AAVS1 targeting, the pAAVS1-P-CAG-$mNG2_{1-10}$ repair template was purified prior to nucleofection using the GeneJET plasmid maxiprep kit (Thermo Fisher Scientific). Synthetic sgRNAs were purchased from MilliporeSigma and Synthego (listed in *Supplementary file 3*) and ArciTect Cas9-eGFP and SpCas9-2NLS were purchased from STEMCELL Technologies and Synthego, respectively. sgRNA spacer sequences were obtained from previous studies (listed in *Supplementary file 3*) or designed using Benchling (*Doench et al., 2016*; *Hsu et al., 2013*). Cas9/sgRNA complexes were pre-mixed at room temperature for 15 min before nucleofection. ssODN repair templates were purchased from BioCorp and Thermo Fisher Scientific (listed in *Supplementary file 4*). For AAVS1 targeting, 201B7 cells were transfected with 1 µg of repair template and 7.5 pmoles sgRNA pre-complexed with 7.5 pmoles Cas9. For endogenous tagging with $mNG2_{11}$, cells were transfected with 82 pmoles ssODN and 91.5 pmoles sgRNA pre-complexed with 30.5 pmoles Cas9. For transient H2B-$mNG2_{11}$ expression, cells were transfected with 750 ng of plasmid purified using the GeneJET plasmid miniprep kit (Thermo Fisher Scientific). Briefly, $5x10^5$ cells were resuspended in 20 µL P3 nucleofection buffer, mixed with the transfection reagents, and nucleofected using the DN100 program on a 4D-Nucleofector (Lonza). After nucleofection, the cells were gently resuspended in media and transferred into six-well plates in mTeSR Plus media containing CloneR2 reagent (STEMCELL Technologies) for AAVS1 targeting, or 24-well plates containing mTeSR Plus media with 10 µM ROCK inhibitor for $mNG2_{11}$ tagging or transient protein expression.

## Antibiotic selection

For AAVS1 integration, antibiotic selection was carried out as described in *Oceguera-Yanez et al., 2016*. Briefly, cells were left to recover for 3 days following transfection. On day 3, the media was changed to media with 0.5 µg/mL puromycin, which was changed every day for 10 days. Following antibiotic selection, colonies were passaged and allowed to become confluent. Edited populations were frozen and subjected to clonal isolation by FACS.

## Fluorescence-activated cell sorting, single-cell recovery, and flow cytometry

FACS was used to enrich populations of edited cells or for clonal isolation. Cells were sorted using a FACSMelody cell sorter (BD Biosciences) after recovery from antibiotic selection or 8 days after nucleofection for endogenous tagging. Briefly, cells were dissociated using Accutase, resuspended in PBS (Wisent), and then passed through a 35 µm strainer to remove large cell clumps. Cells were sorted using gates set to capture individual fluorescent cells. Individual cells or enriched populations of 500 tagged cells were sorted into individual wells of a 96-well plate containing recovery media [mTeSR Plus media with CloneR2 reagent and Pen-Strep (50 units/mL Penicillin and 50 µg/mL Streptomycin; Wisent)]. To recover clones, cells were kept in recovery media for 6 days with media changes on days 2 and 4. The media was then changed daily with mTeSR Plus until day 11. On day 11, the cells were passaged into fresh 96-well plates and grown to confluency before freezing and screening.

For flow cytometry, cells were prepared and analyzed using the same protocol, and data was analyzed using the R package CytoExploreR (*Hammill, 2021*).

## Screening clones in 96-well plates

96-well plates containing single-cell clones were imaged on a Cytation 5 microscope (Agilent) 10 days after sorting. Briefly, 3 × 4 images were acquired for each well using a 4 X phase contrast objective and stitched into one image per well. Laser autofocus was used to find the focal plane where colonies were expected to be found. Images were exported and analyzed using a custom ImageJ macro designed to identify wells with colonies. Briefly, the colony segmentation macro first crops out the well edge from the images, and identifies colonies by subtracting background signal, performing a Gaussian blur, and thresholding high-contrast regions. Finally, it outlines objects and overlays them with the original well image for quality control. The macro also determines the object number to exclude wells with debris and/or multiple colonies and provides a list of positive and negative wells. The code for this macro is available on GitHub (https://github.com/CMCI/colony_screening, copy archived at *CMCI, 2023*).

## Cell lysis for PCR

Cell lysates were generated for all PCR-based assays. Edited clones grown in 96-well plates were dissociated with Accutase and split into two 96-well plates for freezing and cell lysis. The cells were washed once with PBS and resuspended thoroughly in 50 µL of QuickExtract reagent (Lucigen) per well. The resuspended cells were then transferred to PCR-compatible 96-well plates and placed at 65 °C for 15 min followed by brief vortexing, then 98 °C for 15 min, followed by 4 °C. Lysates were stored at –20 °C and thawed as needed for experiments.

## qPCR-based screening

qPCR was performed on the ViiA7 Real-Time PCR system (Applied Biosystems). The VIC-labeled RNaseP assay was purchased from Thermo Fisher Scientific and used as an internal PCR control. The AmpR-specific primers and hydrolysis probe were taken from *Roberts et al., 2017*, and the $mNG2_{1\text{-}10}$-specific primers and hydrolysis probe were designed as follows:

> Forward primer: 5'-TACCGCTACACCTACGAGGG-3'
> Reverse primer: 5'- GTCATCACAGGACCGTCAGC-3'
> Probe: 5'–6-FAM/AT CAA AGG A/ZEN/G AGG CCC AGG TGA TG/IABkFQ-3'

The primers and hydrolysis probes were purchased from Thermo Fisher Scientific and IDT, respectively. The AmpR and $mNG2_{1\text{-}10}$ assays were prepared by mixing 18 µM of each primer with 5 µM of the hydrolysis probe. qPCR reactions consisted of 1 µL cell lysate, 0.5 µL RNaseP assay (Thermo Fisher Scientific), 0.5 µL $mNG2_{1\text{-}10}$ or AmpR assay, 5 µL QuantStudio 3D Digital PCR Master Mix V2 (Thermo Fisher Scientific), and 3 µL water for a final volume of 10 µL. The qPCR thermocycling conditions were carried out as per the manufacturer's instructions for the QuantStudio 3D Digital PCR Master Mix V2. The data was analyzed using the ViiA7 software and clones that were positive for $mNG2_{1\text{-}10}$ and negative for AmpR were selected for further screening by PCR as described below.

## PCR screening and sequencing

Integration of the mNG2$_{1-10}$ cassette at the AAVS1 locus was verified by amplifying the left and right junctions of the integration, as well as the WT AAVS1 locus. Briefly, 50 µL PCR reactions were prepared as follows: 10 µL GC buffer (Thermo Fisher Scientific), 1 µL 10 mM dNTP mix (Thermo Fisher Scientific), 1 µL DMSO (Thermo Fisher Scientific), 0.25 µL of each primer at 100 µM, 5 µL cell lysate, 0.5 µL Phusion polymerase (Thermo Fisher Scientific) and 32 µL water. The primers used for each PCR reaction are listed in *Supplementary file 5*. The following touchdown cycles were then performed: 98 °C for 3 min, followed by 40 cycles of: 98 °C for 10 s, initial annealing at 72 °C and decreasing by 1 °C every cycle until down to 55 °C, and 72 °C for 30 s per kb; followed by a final extension at 72 °C for 10 min and hold at 12 °C. The PCR products were run on a 0.8% agarose gel stained with ethidium bromide and analyzed manually. The bands corresponding to the expected amplicons were gel extracted using the GeneJET gel extraction kit (Thermo Fisher Scientific) and sequenced by Sanger sequencing (Eurofins).

## Digital PCR

Digital PCR reactions were prepared as follows: 1.5 µL cell lysate, 8.7 µL QuantStudio 3D Digital PCR Master Mix V2 (Thermo Fisher Scientific), 0.87 µL RNaseP reference assay (Thermo Fisher Scientific), 0.87 µL mNG2$_{1-10}$ or AmpR assay (prepared as described above), and 5.46 µL water. The amount of lysate was calculated based on the number of cells used and the dynamic range of the QuantStudio 3D system (400–4000 copies/µL), and input lysate volumes were adjusted as needed to obtain data within this dynamic range. 14.5 µL of the reaction mix was loaded onto dPCR chips using the QuantStudio 3D Digital PCR 20 K Chip Kit V2 (Thermo Fisher Scientific), taken through PCR thermocycling and analyzed on the QuantStudio 3D system (Thermo Fisher Scientific) according to manufacturer's instructions. Data was analyzed using the QuantStudio 3D AnalysisSuite Cloud Software (Thermo Fisher Scientific).

## Karyotyping

G-banding karyotyping was carried out and analyzed by the Banque de cellules leucémiques du Québec. 22 cells in metaphase were analyzed at a resolution of 400 bands per haploid karyotype and showed normal karyotypes (46, XX).

## Pluripotency marker staining

Cells were fixed and stained for common pluripotency markers to verify the absence of differentiation. smNG2-P cells were dissociated with Accutase, washed once with PBS, and fixed in 4% paraformaldehyde (PFA) in PBS for 30 min at room temperature. To stain OCT3/4 and NANOG, the cells were washed once with PBS and treated with a permeabilization/blocking solution containing 0.1% Triton X-100 (Sigma) and 5% Normal Donkey Serum (NDS; Jackson ImmunoResearch) in PBS for 30 min at room temperature. To stain TRA-1–60, which is a cell surface marker, the cells were washed once and treated with a blocking solution containing 5% NDS in PBS for 30 min at room temperature. The following antibody dilutions were prepared in permeabilization/blocking solution or blocking solution: 1:25 anti-NANOG antibody (1 µg/mL final concentration, PCRP-NANOGP1-2D8, DSHB), 1:20 anti-OCT3/4 antibody (10 µg/mL final concentration; Santa Cruz Biotechnology), or 1:20 Alexa488-conjugated anti-TRA-1–60 antibody (7.5 µg/mL final concentration; STEMCELL Technologies). The cells were stained in 100 µL of diluted antibody per 10$^6$ cells overnight at 4 °C. For unconjugated primary antibodies, the cells were washed with permeabilization/blocking solution and stained with a 1:400 dilution of Alexa488-conjugated anti-mouse antibody (Invitrogen) for 1 hr at 4 °C. The cells were then washed with permeabilization/blocking solution or blocking solution and resuspended in 1 mL PBS before flow cytometry. Cells stained with only the secondary antibody were used as a negative control.

## Trilineage differentiation and differentiation marker staining

Cells were differentiated into the three germ layers to ensure they retained pluripotent potential after editing. Directed differentiation was performed using the STEMdiff Trilineage Differentiation Kit (STEMCELL Technologies) as per the manufacturer's instructions. The following modifications were made: ectoderm differentiation was carried out on 24-well plates coated with iMatrix-511 Silk by seeding the recommended number of cells (400,000) on day 0, mesoderm differentiation was carried

out on 24-well plates coated with iMatrix-511 Silk by seeding 50,000 cells per well on day 0, and endoderm differentiation was carried out on 24-well plates coated with Matrigel (Corning) by seeding 20,000 cells per well on day 0. After differentiation, the cells were harvested by dissociation with Accutase, washed with PBS, fixed with PFA, and stained as described above for PAX6 (ectoderm lineage), FOXA2 (endoderm lineage), or the cell surface marker NCAM (mesoderm lineage). The following antibody dilutions were prepared in permeabilization/blocking solution or blocking solution: 1:45 anti-PAX6 antibody for ectoderm cells (1 µg/mL final concentration, PAX6, DSHB), 1:240 anti-NCAM antibody for mesoderm cells (0.25 µg/mL final concentration, 5.1H11, DSHB), 1:50 PE-conjugated anti-FOXA2 antibody (1 µg/mL final concentration; BD Biosciences) and 1:400 dilution of Alexa488-conjugated anti-mouse antibody (Invitrogen). The cells were stained in 100 µL of diluted antibody per $10^6$ cells for 1 hr at 4 °C. Stained undifferentiated cells and unstained cells were used as negative controls.

## Off-target sequencing

Potential off-target sites for the AAVS1 sgRNA were selected from *Wang et al., 2014*, Benchling (*Hsu et al., 2013*), and CRISPOR (*Concordet and Haeussler, 2018*) based on previous studies of Cas9 or predicted off-target sites with high scores. Each off-target site was amplified as described above by touchdown Phusion PCR from 201B7 and smNG2-P cell lysates, and sequenced by Sanger sequencing (Eurofins). The primers used to amplify the 12 off-target sites were designed using Primer-BLAST (*Ye et al., 2012*) and are listed in *Supplementary file 5*. No mutations were found at the predicted off-target cut sites, and all 12 sites sequenced in the edited cell line matched the sequence from the WT cells.

## Mycoplasma test

PCR-based mycoplasma detection was performed using the Mycoplasma PCR detection kit (Applied Biological Materials) according to the manufacturer's protocol. For each test, a positive and negative control were included. The PCR products were run on a 0.8% agarose gel stained with ethidium bromide and analyzed manually.

## Nanopore sequencing and data analysis

For Nanopore sequencing of mixed edited populations, PCR reactions were carried out as described above with a stable annealing temperature of 65 °C for 30 cycles to minimize non-specific amplification and PCR bias. The samples were then gel extracted and pooled in equimolar amounts before library preparation for Nanopore sequencing.

For multiplexed clone sequencing, the edited loci from each clone were amplified individually and barcoded by PCR prior to Nanopore sequencing. For this, individual clones were subjected to a first round of PCR to amplify the target loci and add universal adapters with the primers listed in *Supplementary file 5*. The locus-specific primers were designed using Primer-BLAST (*Ye et al., 2012*), and the sequence of the universal adapters were taken from *Karst et al., 2021*. The first round of PCR was carried out as described above with annealing at 65 °C and for 15 cycles. The PCR products were diluted 1/100 into the second PCR reactions, which contained a unique pair of barcoding primers for each clone for a specific target locus. The second PCR was carried out as described above with annealing at 65 °C and for 15 cycles. The PCR products for a specific edited locus were pooled, run on a 0.8% agarose gel, and gel extracted. The concentration of pooled products was then measured and the PCR products for clones edited at different target loci were then pooled in equal molar ratios before Nanopore library preparation.

Library preparation for Nanopore sequencing was carried out using the NEBNext Companion Module for Oxford Nanopore Technologies Ligation Sequencing (New England Biolabs), the Ligation Sequencing Kit V14, and the Flongle Sequencing Expansion (Oxford Nanopore Technologies), as per the manufacturer's protocols. The final library concentration was measured using a Qubit fluorometer (Invitrogen) using the Qubit dsDNA HS assay kit (Invitrogen), and 5 fmoles of the library were loaded onto a Flongle flow cell and sequenced overnight.

Basecalling of the Nanopore FAST5 files was performed with Guppy (R10.4.1 flow cell, 400 bps, super accuracy configuration) using the Compute Canada server, and low-quality reads with Q scores below 10 were excluded. Reads were demultiplexed using a custom Bowtie-based pipeline (obtained

from Daniel Giguère, Flow Genomics, 2022, available at https://github.com/frba/nanopore_demulti-plex [*Araujo, 2022*]). When sequencing amplicons from mixed edited populations, reads were demultiplexed based on the presence of either of two gene-specific barcodes (listed in *Supplementary file 6*). The FASTQ files were analyzed using CRISPResso2 (*Clement et al., 2019*) to obtain the frequency of WT, HDR, and indel alleles in the population. All HDR alleles were included in the same category because of the high sequencing error rate. For clone sequencing, reads were demultiplexed based on the presence of two gene-specific and two clone-specific barcodes (listed in *Supplementary file 6*). The FASTQ reads were aligned to the expected WT and edited sequences using Minimap2 (*Li, 2018*), and the alignments were visualized using IGV (*Thorvaldsdóttir et al., 2013*). The genotypes of edited clones were inferred manually by looking at the alignments of sequencing reads with the WT and edited alleles for each clone.

## Fluorescence microscopy

The fluorescence from $mNG2_{1-10}/mNG2_{11}$ complementation was visualized the day after transfection with the $pH2B-mNG2_{11}$ plasmid using a Cytation 5 microscope (Agilent) equipped with a 20 x/0.45 NA phase contrast objective, a 465 nm LED cube and green filter cube (excitation 469/35 nm, emission 525/39 nm and dichroic mirror 497 nm).

Cells with $mNG2_{11}$ integrated at endogenous loci were imaged at least 8 days after nucleofection on a Leica DMI6000B inverted epifluorescence microscope equipped with a EL6000 mercury lamp and a GFP3035B filter cube (excitation 472/30 nm, emission 520/35 and dichroic mirror 495 nm) using a 20 x/0.35 NA objective, an Orca R2 CCD camera (Hamamatsu) and Volocity software (PerkinElmer).

Because of its weak fluorescent signal, the $CDH1-mNG2_{11}$ edited cells were seeded in a 35 mm polymer coverslip dish (Cellvis) coated with iMatrix-511 Silk, and imaged on an inverted Nikon Eclipse Ti microscope (Nikon) equipped with a Livescan Sweptfield scanner (Nikon), Piezo Z stage (Prior), IXON 879 EMCCD camera (Andor), and a 488 nm laser (50 mW, Agilent) using the 100 x/1.45 NA objective.

For live imaging, cells were adapted to culture on Matrigel (Corning) for at least 2 passages. Cells were passaged as described above, and seeded onto 35 mm polymer coverslip dishes (Cellvis) freshly coated with Matrigel. The cells were then allowed to grow into colonies for 4–5 days before imaging. Fresh mTeSR Plus media was added to the cells at least 30 min before imaging. To visualize chromatin, Hoechst 33342 (Invitrogen) was added to the cells at a final concentration of 1.78 µM (1 µg/mL) for 30 min prior to imaging. Imaging was performed using the inverted Nikon Eclipse Ti microscope with the Livescan Sweptfield scanner described above, equipped with 405 and 488 nm lasers (50 mW, Agilent). The cells were kept at 37 °C and 5% $CO_2$ during imaging in an INU-TiZ-F1 chamber (MadCityLabs). Z-stacks of 9 slices at 1 µm intervals were acquired every minute using NIS Elements software (Version 4.0, Nikon). For CARE training, images were collected using low- and high-exposure settings with a 16-slice Z-stack of 0.75 µm intervals. The imaging parameters (laser power and exposure time) used for each cell line in low- and high-exposure conditions are listed in *Supplementary file 7*.

## Image restoration by CARE

For image restoration, we used the CSBDeep neural network developed by *Weigert et al., 2018*. Briefly, a training dataset composed of >100 matched two-channel fluorescent images (green for mNG and blue for Hoechst) with low- and high-exposure settings was acquired for each tagged cell line. The training datasets were designed to include cells in interphase and at different stages of mitosis. We used the Python implementation of CSBDeep to train the standard model of the neural network for each cell line individually. We then applied this model to restore timelapse Z-stack two-channel images of the corresponding cell lines.

## Image analysis

All images acquired using NIS Elements (Nikon) were opened in Fiji (Version 2.3, NIH) for analysis. For signal-to-noise ratio measurements, two regions of interest were drawn over an area of homogeneous signal inside a cell, and over a region of background signal (in a region with no cells). The mean pixel intensity and standard deviation (SD) were measured for both regions of interest, and the signal-to-noise ratio was calculated as follows: SNR = (mean signal −mean background)/signal SD. The

measurement was repeated using the same regions of interest for matched low- and high-exposure images.

Linescans were performed and measured for tagged cell lines using a macro in Fiji modified from *Ozugergin et al., 2022*. The macro was designed to isolate the 488 nm channel from the image file, subtract background signal, and perform a bleach correction. The desired timepoint and two central Z slices were picked manually, and the macro generated a Z-stack average projection. A five-pixel-wide line was then traced along the cortex of the cell, from one pole to the other, along with a straight one-pixel-wide line to define the midplane. The macro then measured the fluorescence intensity of each pixel along the length of the linescan and positioned the pixels in relation to the midplane. For the measurement of tubulin at the central spindle, the same macro was used to draw a five-pixel-wide line across the cell equator. All data was exported for use in Excel (Microsoft) and Prism (Version 9.3, GraphPad) for further analysis.

For breadth measurements, the number of pixels above 50% of the normalized peak intensities were counted for each linescan and converted to microns. Pixels with intensities higher than the cutoff value outside of the peak region were excluded from these calculations. For measurements of the ratio of cortical to cytosolic protein in metaphase cells, the average intensity of pixels at the cortex was measured by a linescan as described above, and the average intensity of pixels in the cytosol was measured by drawing a region of interest over the cytosol.

## Statistical analysis

Box and whiskers plots were generated using Prism (Version 9.3, GraphPad) to show median values (central line), quartiles (box edges), and minimum and maximum values (whiskers). Statistical significance was tested using a Brown-Forsythe and Welch's ANOVA, followed by multiple comparisons using Dunnett's T3 test, or by Welch's t test (Graphpad Prism version 9.3). Significance levels were defined as: $p > 0.05$ non-significant (ns), $*p \leq 0.05$; $**p \leq 0.01$; $***p \leq 0.001$; $****p \leq 0.0001$.

## Acknowledgements

We thank Nicholas D Gold and Angela Quach from the Concordia Genome Foundry for their help with equipment use and analysis of the Nanopore sequencing data. We also thank the Concordia Center for Microscopy and Cellular Imaging for help with imaging and analysis. We thank Dr. Knut Woltjen from the Center for iPS Cell Research and Application (CiRA) and his lab for guidance with iPS culture and editing, and researchers from the National Research Council of Canada (NRC) for feedback on this project. This research was funded by the NRC Disruptive Technology Solutions for Cell and Gene Therapy Challenge program and the NSERC Discovery Grant RGPIN-2023–04805. MCH was supported by a FRQNT B2X and NSERC CREATE SynBioApps scholarships. NPP was supported by a FRQNT B2X scholarships. AJP was supported by Concordia University Research Chair in Cancer Cell Biology and VJJM was supported by the Applied Synthetic Biology Senior Research Chair.

## Additional information

### Funding

| Funder | Grant reference number | Author |
| --- | --- | --- |
| Fonds de recherche du Québec – Nature et technologies | Doctoral Training Scholarship B2X | Mathieu C Husser Nhat Pham |
| Concordia University | SynBioApps scholarship | Mathieu C Husser |
| National Research Council Canada | Disruptive Technology Solutions for Cell and Gene Therapy Challenge | Vincent JJ Martin |
| Natural Sciences and Engineering Research Council of Canada | Discovery Grant | Alisa Piekny |

| Funder | Grant reference number | Author |
| --- | --- | --- |
| Natural Sciences and Engineering Research Council of Canada | RGPIN-2023-04805 | Alisa Piekny |

The funders had no role in study design, data collection and interpretation, or the decision to submit the work for publication.

## Author contributions

Mathieu C Husser, Conceptualization, Data curation, Software, Formal analysis, Validation, Investigation, Visualization, Methodology, Writing – original draft, Writing – review and editing; Nhat P Pham, Data curation, Investigation, Methodology; Chris Law, Data curation, Software, Validation, Methodology, Writing – original draft; Flavia RB Araujo, Software, Methodology; Vincent JJ Martin, Conceptualization, Resources, Supervision, Funding acquisition, Investigation, Methodology, Writing – original draft, Project administration, Writing – review and editing; Alisa Piekny, Conceptualization, Supervision, Funding acquisition, Investigation, Visualization, Methodology, Writing – original draft, Project administration, Writing – review and editing

## Author ORCIDs

Mathieu C Husser ⓘ http://orcid.org/0000-0001-6925-6440
Vincent JJ Martin ⓘ https://orcid.org/0000-0001-7511-115X
Alisa Piekny ⓘ https://orcid.org/0000-0002-4264-6980

Reviewer #1 (Public Review): https://doi.org/10.7554/eLife.92819.3.sa1
Reviewer #2 (Public Review): https://doi.org/10.7554/eLife.92819.3.sa2
Reviewer #3 (Public Review): https://doi.org/10.7554/eLife.92819.3.sa3
Author response https://doi.org/10.7554/eLife.92819.3.sa4

# Additional files

## Supplementary files

• Supplementary file 1. Characterization and validation of the smNG2-P cell line. The results from the validation and quality control steps carried out on the smNG2-P cell line are listed.

• Supplementary file 2. AAVS1 off-target sites sequencing in 201B7 smNG2-P cells. The top off-target sites for the AAVS1 sgRNA are listed, along with the results of sequencing these sites in the split mNeonGreen2 parental (smNG2-P) cell line compared to wild-type (WT) cells.

• Supplementary file 3. List of sgRNAs used in this study. The sequences of all sgRNAs used in this study are listed.

• Supplementary file 4. List of single-stranded repair templates (ssODNs) used for endogenous tagging with mNG2$_{11}$. The sequences of the single-stranded oligonucleotides used as repair templates to integrate mNG2$_{11}$ at various loci are shown. For each oligonucleotide, the homology arms are shown in blue, the mNG2$_{11}$ tag in green, the protein linker in purple and mutations designed to remove homology and prevent Cas9 re-cutting in red.

• Supplementary file 5. List of genotyping primers. The primers used to amplify and sequence each edited locus are listed.

• Supplementary file 6. List of barcoding primers. The primers used to add unique barcodes onto locus-specific amplicons for pooled Nanopore sequencing are shown.

• Supplementary file 7. List of live imaging parameters. For each cell lines imaged, the laser power and exposure time used to acquire low- and high-exposure images are listed. These parameters were used to acquire the neural network training dataset, and the low-exposure settings were used to acquire timelapse images for restoration.

• MDAR checklist

## Data availability

Source data for each figure are provided in the corresponding Source Data files. RNA-seq data from *Iwasaki et al., 2022* was obtained from the Gene Expression Omnibus (GEO) Database (Accession number GSE199820).

The following previously published dataset was used:

| Author(s) | Year | Dataset title | Dataset URL | Database and Identifier |
|---|---|---|---|---|
| Iwasaki M, Takahashi K, Okubo C | 2022 | Post-transcriptionally regulated genes are essential for pluripotent stem cell survival | https://www.ncbi.nlm.nih.gov/geo/query/acc.cgi?acc=GSE199820 | NCBI Gene Expression Omnibus, GSE199820 |

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
