## [Editor Report · eLife assessment]

In this study, the authors develop a strategy for fluorophore-tagging endogenous proteins in human induced pluripotent stem cells (iPSCs) using a split mNeonGreen approach, and they conclude that the system will be appropriate for performing live imaging studies of highly dynamic cellular processes such as cytokinesis in iPSCs. Experimentally, the methods are **solid**, and the data presented support the authors' conclusions. Overall, these methodologies should be **useful** to a wide audience of cell biologists who want to study protein localization and dynamics at endogenous levels in iPSCs.

---

## [Referee Report · Reviewer #1 (Public Review)]

Summary:

In this manuscript the authors have applied an asymmetric split mNeonGreen2 (mNG2) system to human iPSCs. By integrating a constitutively expressed long fragment of mNG2 at the AAVS1 locus, this allows other proteins to be tagged through the use of available ssODN donors. This removes the need to generate long AAV donors for tagging, thus greatly facilitating high-throughput tagging efforts. The authors then demonstrate the feasibility of the method by successfully tagging 9 markers expressed in iPSC at various, and one expressed upon endoderm differentiation. Several additional differentiation markers were also successfully tagged but not subsequently tested for expression/visibility. As one might expect for high-throughput tagging, a few proteins, while successfully tagged at the genomic level, failed to be visible. Finally, to demonstrate the utility of the tagged cells, the authors isolated clones with genes relevant to cytokinesis tagged, and together with an AI to enhance signal to noise ratios, monitored their localization over cell division.

Strengths

Reviewer Comment: Characterization of the mNG2 tagged parental iPSC line was well and carefully done including validation of a single integration, the presence of markers for continued pluripotency, selected off-target analysis and G-banding-based structural rearrangement detection.

The ability to tag proteins with simple ssODNs in iPSC capable of multi-lineage differentiation will undoubtedly be useful for localization tracking and reporter line generation.

Validation of clone genotypes was carefully performed and highlights the continued need for caution with regards to editing outcomes.

Weaknesses

Reviewer Comment: IF and flow cytometry figures lack quantification and information on replication. How consistent is the brightness and localization of the markers? How representative are the specific images? Stability is mentioned in the text but data on the stability of expression/brightness is not shown.

Author Response: To address this comment, we have quantified the mean fluorescence intensity of the tagged cell populations in Fig. S3B-T. This data correlates well with the expected expression levels of each gene relative to the others (Fig. S3A), apart from CDH1 and RACGAP1, which are described in the discussion.

Reviewer Reply: Great, thanks.

Reviewer Comment: The localization of markers, while consistent with expectations, is not validated by a second technique such as antibody staining, and in many cases not even with Hoechst to show nuclear vs cytoplasmic.

Author Response: We find that the localization of each protein is distinct and consistent with previous studies. To address this comment, we have added an overlay of the green fluorescence images with brightfield images to better show the location of the tagged protein relative to the nuclei and cytoplasm. We have also added references to other studies that showed the same localization patterns for these proteins in iPSCs and other relevant cell lines.

Reviewer Reply: There was no question that the localization fit with expectations, however, this still doesn't show that in the same cell the tag is in the same spot. It would have been fairly simple to do for at least a handful of markers, image, fix and stain to demonstrate unequivocally the tag and protein are co-localized. Of course, this isn't damning by any means, it just would have been nice.

Reviewer Comment: For the multi-germ layer differentiation validation, NCAM is also expressed by ectoderm, so isn't a good solo marker for mesoderm as it was used. Indeed, the kit used for the differentiation suggests Brachyury combined with either NCAM or CXCR4, not NCAM alone.

Author Response: Since Brachyury is the most common mesodermal marker, we first tested differentiation using anti-Brachyury antibodies, but they did not work well for flow cytometry. We then switched to anti-NCAM antibodies. Since we used a kit for directed differentiation of iPSCs into the mesodermal lineage, NCAM staining should still report for successful differentiation. In the context of mixed differentiation experiments (embryoid body formation or teratoma assay), NCAM would not differentiate between ectoderm and mesoderm. The parental cells (201B7) have also been edited at the AAVS1 locus in multiple other studies, with no effect on their differentiation potential.

Reviewer Reply: This is placing a lot of trust in the kit that it only makes what it says it makes. It could have been measured by options other than flow such as qPCR, Western blot, or imaging, but fine.

Reviewer Comment: Only a single female parental line has been generated and characterized. It would have been useful to have several lines and both male and female to allow sex differences to be explored.

Author Response: We agree that it would be interesting (and important) to study differences in protein localization between female and male cell types, and from different individuals with different genetic backgrounds. We see our tool as opening a door for cell biology to move away from randomly collected, transformed, differentiated cell types to more directed comparative studies of distinct normal cell types. Since few studies of cell biological processes have been done in normal cells, a first step is to understand how processes compare in an isogenic background, then future studies can reveal how they compare with other individuals and sexes. We hope that either our group or others will continue to build similar lines so that these studies can be done.

Reviewer Reply: Fair enough.

Reviewer Comment: The AI-based signal to noise enhancement needs more details and testing. Such models can introduce strong assumptions and thus artefacts into the resolved data. Was the model trained on all markers or were multiple models trained on a single marker each? For example, if trained to enhance a single marker (or co-localized group of markers), it could introduce artefacts where it forces signal localization to those areas even for others. What happens if you feed in images with scrambled pixel locations, does it still say the structures are where the training data says they should be? What about markers with different localization from the training set. If you feed those in, does it force them to the location expected by the training data or does it retain their differential true localization and simply enhance the signal?

Author Response: The image restoration neural network was used as in Weigert et al., 2018. The model was trained independently for each marker. Each trained model was used only on the corresponding marker and with the same imaging conditions as the training images. From visual inspection, the fluorescent signal in the restored images was consistent with the signal in the raw images, both for interphase and mitotic cells. We found very few artefacts of the restoration (small bright or dark areas) that were discarded. We did not try to restore scrambled images or images of mismatched markers.

Reviewer Reply: I understand. What I'm saying is that for the restoration technique to be useful you need to know that it won't introduce artefacts if you have an unexpected localization. Think of it this way, if you already know the localization, then there's no point measuring it. If you don't, or there's a possibility that it is somewhere unexpected, then you need to know with confidence that your algorithm will be able to accurately detect that unexpected localization. As such, it would be extremely important to validate that your restoration algorithm will not bias the results to the expected localization if the true localization is unexpected/not seen in the training dataset. It would have been extremely trivial to run this analysis and I do not feel this comment has been in any way adequately addressed.

---

## [Referee Report · Reviewer #2 (Public Review)]

Summary:

The authors have generated human iPSC cells constitutively expressing the mNG21-10 and tested them by endogenous tagging multiple genes with mNG211 (several tagged iPS cell lines clones were isolated). With this tool they have explored several weakly expressed cytokinesis genes gained insights into how cytokinesis occurs.

Strengths:

(i) Human iPSC cells are used

Weaknesses:

(i) The manuscript is extremely incremental, no improvements are present in the split-Fluorescent (split-FP) protein variant used nor in the approach for endogenous tagging with split-FPs (both of them are already very well established and used in literature as well as in different cell types).

(ii) The fluorescence intensity of the split mNeonGreen appears rather low, for example in Figure 2C the H2BC11, ANLN, SOX2 and TUBB3 signals are very noisy (differences between the structures observed are almost absent). For low expression targets this is an important limitation. This is also stated by the authors but image restoration could not be the best solution since a lot of biologically relevant information will be lost anyway.

(iii) there is no comparison with other existing split-FP variants, methods, or imaging and it is unclear what the advantages of the system are.

---

## [Referee Report · Reviewer #3 (Public Review)]

The authors report on the engineering of an induced Pluripotent Stem Cell (iPSC) line that harbours a single copy of a split mNeonGreen, mNG2(1-10). This cell line is subsequently used to take endogenous protein with a smaller part of mNeonGreen, mNG2(11), enabling complementation of mNG into a fluorescent protein that is then used to visualize the protein. The parental cell is validated and used to construct several iPSC line with endogenously tagged proteins. These are used to visualize and quantify endogenous protein localisation during mitosis.

I see the advantage of tagging endogenous loci with small fragments, but the complementation strategy has disadvantages that deserve some attention. One potential issue is the level of the mNG2(1-10). In addition, this may probably not work for organelle-resident proteins, where the mNG2(11) tag is localised in a membrane enclosed compartment.

Overall the tools and resources reported in this paper will be valuable for the community that aims to study proteins at endogenous levels.

---

## [Author Response]

The following is the authors’ response to the original reviews.

**eLife assessment**
In this study, the authors develop a useful strategy for fluorophore-tagging endogenous proteins in human induced pluripotent stem cells (iPSCs) using a split mNeonGreen approach. Experimentally, the methods are solid, and the data presented support the author's conclusions. Overall, these methodologies should be useful to a wide audience of cell biologists who want to study protein localization and dynamics at endogenous levels in iPSCs.
**Public Reviews:**

**Reviewer #1 (Public Review):**
Summary:In this manuscript, the authors have applied an asymmetric split mNeonGreen2 (mNG2) system to human iPSCs. Integrating a constitutively expressed long fragment of mNG2 at the AAVS1 locus, allows other proteins to be tagged through the use of available ssODN donors. This removes the need to generate long AAV donors for tagging, thus greatly facilitating high-throughput tagging efforts. The authors then demonstrate the feasibility of the method by successfully tagging 9 markers expressed in iPSC at various, and one expressed upon endoderm differentiation. Several additional differentiation markers were also successfully tagged but not subsequently tested for expression/visibility. As one might expect for high-throughput tagging, a few proteins, while successfully tagged at the genomic level, failed to be visible. Finally, to demonstrate the utility of the tagged cells, the authors isolated clones with genes relevant to cytokinesis tagged, and together with an AI to enhance signal-to-noise ratios, monitored their localization over cell division.Strengths:Characterization of the mNG2 tagged parental iPSC line was well and carefully done including validation of a single integration, the presence of markers for continued pluripotency, selected offtarget analysis, and G-banding-based structural rearrangement detection.The ability to tag proteins with simple ssODNs in iPSC capable of multi-lineage differentiation will undoubtedly be useful for localization tracking and reporter line generation.Validation of clone genotypes was carefully performed and highlights the continued need for caution with regard to editing outcomes.Weaknesses:IF and flow cytometry figures lack quantification and information on replication. How consistent is the brightness and localization of the markers? How representative are the specific images? Stability is mentioned in the text but data on the stability of expression/brightness is not shown.

To address this comment, we have quantified the mean fluorescence intensity of the tagged cell populations in Fig. S3B-T. This data correlates well with the expected expression levels of each gene relative to the others (Fig. S3A), apart from CDH1 and RACGAP1, which are described in the discussion.

The images in Fig. 2 show tagged populations enriched by FACS so they are non-clonal and are representative of the diversity of the population of tagged cells.

The images shown in Fig. 3 are representative of the clonal tagged populations. The stability of the tag was not quantified directly. However, the fluorescence intensity was very stable across cells in clonal populations. Since these populations were recovered from a single cell and grown for several weeks, this low variability across cells in a population suggests that these tags are stable.

The localization of markers, while consistent with expectations, is not validated by a second technique such as antibody staining, and in many cases not even with Hoechst to show nuclear vs cytoplasmic.

We find that the localization of each protein is distinct and consistent with previous studies. To address this comment, we have added an overlay of the green fluorescence images with brightfield images to better show the location of the tagged protein relative to the nuclei and cytoplasm. We have also added references to other studies that showed the same localization patterns for these proteins in iPSCs and other relevant cell lines.

For the multi-germ layer differentiation validation, NCAM is also expressed by ectoderm, so isn't a good solo marker for mesoderm as it was used. Indeed, the kit used for the differentiation suggests Brachyury combined with either NCAM or CXCR4, not NCAM alone.

Since Brachyury is the most common mesodermal marker, we first tested differentiation using anti-Brachyury antibodies, but they did not work well for flow cytometry. We then switched to anti-NCAM antibodies. Since we used a kit for directed differentiation of iPSCs into the mesodermal lineage, NCAM staining should still report for successful differentiation. In the context of mixed differentiation experiments (embryoid body formation or teratoma assay), NCAM would not differentiate between ectoderm and mesoderm. The parental cells (201B7) have also been edited at the AAVS1 locus in multiple other studies, with no effect on their differentiation potential.

Only a single female parental line has been generated and characterized. It would have been useful to have several lines and both male and female to allow sex differences to be explored.

We agree that it would be interesting (and important) to study differences in protein localization between female and male cell types, and from different individuals with different genetic backgrounds. We see our tool as opening a door for cell biology to move away from randomly collected, transformed, differentiated cell types to more directed comparative studies of distinct normal cell types. Since few studies of cell biological processes have been done in normal cells, a first step is to understand how processes compare in an isogenic background, then future studies can reveal how they compare with other individuals and sexes. We hope that either our group or others will continue to build similar lines so that these studies can be done.

The AI-based signal-to-noise enhancement needs more details and testing. Such models can introduce strong assumptions and thus artefacts into the resolved data. Was the model trained on all markers or were multiple models trained on a single marker each? For example, if trained to enhance a single marker (or co-localized group of markers), it could introduce artefacts where it forces signal localization to those areas even for others. What happens if you feed in images with scrambled pixel locations, does it still say the structures are where the training data says they should be? What about markers with different localization from the training set? If you feed those in, does it force them to the location expected by the training data or does it retain their differential true localization and simply enhance the signal?

The image restoration neural network was used as in Weigert et al., 2018. The model was trained independently for each marker. Each trained model was used only on the corresponding marker and with the same imaging conditions as the training images. From visual inspection, the fluorescent signal in the restored images was consistent with the signal in the raw images, both for interphase and mitotic cells. We found very few artefacts of the restoration (small bright or dark areas) that were discarded. We did not try to restore scrambled images or images of mismatched markers.

**Reviewer #2 (Public Review):**
Summary:The authors have generated human iPSC cells constitutively expressing the mNG21-10 and tested them by endogenous tagging multiple genes with mNG211 (several tagged iPS cell lines clones were isolated). With this tool, they have explored several weakly expressed cytokinesis genes and gained insights into how cytokinesis occurs.Strengths:Human iPSC cells are used.Weaknesses:i) The manuscript is extremely incremental, no improvements are present in the split-fluorescent (split-FP) protein variant used nor in the approach for endogenous tagging with split-FPs (both of them are already very well established and used in literature as well as in different cell types).

Although split fluorescent proteins and the endogenous tagging methodology had been developed previously, their use in human stem cells has not been explored. We argue that human iPSCs are a valuable model for cell biologists to study cellular processes in differentiating cells in an isogenic context for proper comparison. Many normal human cell types have not been studied at the cellular/subcellular level, and this tool will enable those studies. Importantly, other existing cell lines required transformation to persist in culture and represent a single, differentiated cell type that is not normal. Moreover, the protocols that we developed along with this methodology (e.g. workflows for iPSC clonal isolation that include automated colony screening and Nanopore sequencing) will be useful to other groups undertaking gene editing in human cells. Therefore, we argue that our work opens new doors for future cell biology studies.

ii) The fluorescence intensity of the split mNeonGreen appears rather low, for example in Figure 2C the H2BC11, ANLN, SOX2, and TUBB3 signals are very noisy (differences between the structures observed are almost absent). For low-expression targets, this is an important limitation. This is also stated by the authors but image restoration could not be the best solution since a lot of biologically relevant information will be lost anyway.

The split mNeonGreen tag is one of the brighter fluorescent proteins that is available. The low expression that the reviewer refers to for H2BC11, ANLN, TUBB3 and SOX2 is expected based on their predicted expression levels. Further, these images were taken with cells in dishes using lower resolution imaging and were not intended to be used for quantification. As shown in the images in Figures 3H, when using a different microscope with different optical settings and higher magnification, the localization is very clear and quantifiable without needing to use restoration (e.g., compare H2BC11 and ANLN). Using microscopes with high NA objectives, lasers and EMCCD or sCMOS cameras with high sensitivity can sufficiently detect levels of very weakly expressing proteins that can be quantified above background and compared across cells. It is worth noting that each tag may be studied in very different contexts. For example, ANLN will be useful for studies of cytokinesis, while the loss of SOX2 expression and gain of TUBB3 expression may be used to screen for differentiation rather than for localization per se. The reason for endogenous tagging is to study proteins at their native levels rather than using over-expression or fixation with antibodies where artefacts can be introduced. Endogenous tags tag will also enable studies of dynamic changes in localization during differentiation in an isogenic background as described previously.

Importantly, image restoration is not required to image any of these probes! We use it to demonstrate how a researcher can increase the temporal resolution of imaging weakly-expressed proteins for extended periods of time. This data can be used to compare patterns of localization and reveal how patterns change with time and during differentiation. Imaging with fewer timepoints and altered optical settings will still permit researchers to extract quantifiable information from the raw data without requiring image restoration.

iii) There is no comparison with other existing split-FP variants, methods, or imaging and it is unclear what the advantages of the system are.

We are not sure what the reviewer means by this comment. In the future, we plan to incorporate an additional split-FP variant (e.g., split sfCherry) in this iPSC line to enable the imaging of more than one protein in the same cell. However, the split mNeonGreen system is still amenable for use with dyes with different fluorescence spectra that can mark other cellular components, especially for imaging over shorter timespans. In addition to tagging efficiency, the main advantage of split FPs is its scale, as demonstrated by the OpenCell project by tagging 1,310 proteins endogenously (Cho et al., 2022). We developed protocols that facilitate the identification of edited cell lines with high throughput. We also used multiple imaging methods throughout the study that relied on the use of different microscopes and flow cytometry, demonstrating the flexibility of this tagging system. Even for more weakly expressing proteins, the probe could be sufficiently visualized by multiple systems. Such endogenous tags can be used for everything from simply knowing when cells have differentiated (e.g., loss of SOX2 expression, gain of differentiation markers), to studying biological processes over a range of timescales.

**Reviewer #3 (Public Review):**
The authors report on the engineering of an induced Pluripotent Stem Cell (iPSC) line that harbours a single copy of a split mNeonGreen, mNG2(1-10). This cell line is subsequently used to take endogenous protein with a smaller part of mNeonGreen, mNG2(11), enabling thecomplementation of mNG into a fluorescent protein that is then used to visualize the protein. The parental cell is validated and used to construct several iPSC lines with endogenously tagged proteins. These are used to visualize and quantify endogenous protein localisation during mitosis.I see the advantage of tagging endogenous loci with small fragments, but the complementation strategy has disadvantages that deserve some attention. One potential issue is the level of the mNG2(1-10). Is it clear that the current level is saturating? Based on the data in Figure S3, the expression levels and fluorescence intensity levels show a similar dose-dependency which is reassuring, but not definitive proof that all the mNG2(11)-tagged protein is detected.

We have not quantified the levels of mNG21-10 expression directly. However, the increase in fluorescence observed with highly expressed proteins (e.g., ACTB) supports that mNG21-10 levels must be sufficiently high to permit differences among endogenous proteins with vastly different expression levels. To ensure high expression, we used a previously validated expression system comprised of the CAG promoter integrated at the AAVS1 locus, which has previously been used to provide high and stable transgene expression (e.g. Oceguera-Yanez et al., 2016). We acknowledge that it is difficult to confirm that all of the endogenous mNG211-tagged protein is‘detectable’.

Do the authors see a difference in fluorescence intensity for homo- and heterozygous cell lines that have the same protein tagged with mNG2(11)? One would expect two-fold differences, or not?

To answer this question, we measured the fluorescence intensity of homozygous and heterozygous clones carrying smNG2-anillin and smNG2-RhoA. We found homozygous clones that were approximately twice as bright as the corresponding heterozygous clones (Fig. S4H and I). This suggests that the complementation between mNG21-10 and mNG211 occurs efficiently over a range of mNG211 expression, since anillin is expressed weakly and RhoA is expressed more strongly in iPSCs. However, we also observed some homozygous clones that were not brighter than the corresponding heterozygous clones, which could be due to undetected byproducts of CRISPR or clonal variation in protein expression.

Related to this, would it be favourable to have a homozygous line for expressing mNG2(1-10)?

Our heterozygous cell line leaves the other AAVS1 allele available for integrations of other transgenes for future experiments. While a homozygous line could express more mNG2(1-10), it does not seem to be rate-limiting even with a highly-expressed protein like beta-actin, and we are not sure that it is necessary. The value gained by having the free allele could outweigh the difference in mNG2(1-10) levels.

The complementation seems to work well for the proteins that are tested. Would this also work for secreted (or other organelle-resident) proteins, for which the mNG2(11) tag is localised in a membrane-enclosed compartment?

The interaction between the 1-10 and 11 fragments is strong and should be retained when proteins are secreted. It was recently shown that secreted proteins tagged with GFP11 can be detected when interacting with GFP1-10 in the extracellular space, albeit using over-expression (Minegishi et al., 2023). However, in our work, the mNG21-10 fragment is cytosolic and we have only explored proteins localized to the nucleus or the cytoplasm similar to Cho et al., (2022). By GO annotation, 75% of human proteins are present in the cytoplasm and/or nucleus, which still covers a wide range of proteins of interest. Future versions of our line could include incorporating organelle-targeting peptides to drive the large fragment to specific, non-cytosolic locations.

The authors present a technological advance and it would be great if others could benefit from this as well by having access to the cell lines.

As discussed below, some of the resources are already available, and we are working to make the mNG21-10 cell line available for distribution.

**Recommendations for the authors:**

**Reviewer #2 (Recommendations For The Authors):**
The manuscript is methodological, the main achievement is the generation of a stable iPSC with the split Neon system available for the scientific community. Although it is technically solid, the judgement of this reviewer is that the manuscript should be considered for a more specialised/methodological/resource-based journal.

Indeed, we have submitted this article under the “tools and resources” category of eLife, which publishes methodology-centered papers of high technical quality. We felt this was a good venue for the audience that it can reach compared to more specialized journals that may be more limited in scope. For example, our system will be a useful resource for cell biologists and they are more likely to see it in eLife compared to more specialized journals.

**Reviewer #3 (Recommendations For The Authors):**
(1) The authors present a technological advance and it would be great if others can benefit from this as well. Therefore access to the materials (and data) would be valuable (the authors do a great job by listing all the repair templates and primers).

We have added several pieces of data and information to the supplementary materials, as described below.

For instance:What is the (complete/plasmid) sequence of the AAVS1-mNG2(1-10) repair plasmid? Will it be deposited at Addgene?

The plasmids used in this paper are now available on Addgene, along with their sequences [ID 206042 for pAAVS1-Puro-CAG-mNG2(1-10) and 206043 for pH2B-mNG2(11)].

The ImageJ code for the detection of colonies is interesting and potentially valuable. Will the code be shared (e.g. at Github, or as supplemental text)?

The ImageJ macro has been uploaded to the CMCI Github page(https://github.com/CMCI/colony_screening). The parameters are optimized to perform segmentation on images obtained using a Cytation5 microscope with our specific settings, but they can be tweaked for any other sets of images. The following text has been added to the methods section: “The code for this macro is available on Github(https://github.com/CMCI/colony_screening)”.

The cell line with the mNG2(1-10) as well as other cell lines can be of interest to others. Will the cell lines be made available? If so, can the authors indicate how?

We are in the process of depositing our cell line in a public repository. This process may take some time for quality control. For now, the cells can be made available by requesting them from the corresponding authors.

(2) How well does the ImageJ macro for detection of the colonies in the well work? Is there any comparison of analysis by a human vs. the macro?

In our most recent experiment, the colony screening macro correctly identified 99.5% of wells compared to manual annotation (83/84 positive wells and 108/108 negative wells). For each 96-well plate, imaging takes 25 minutes, and it takes 7 minutes for analysis. Despite a few false negatives, we expect this macro to be useful for large-scale experiments where multiple 96-well plates need to be screened, which would take hours manually.

(3) The CDH labeling was not readily detected by FACS, but was visible by microscopy. Is the labeling potentially disturbed by the procedure (low extracellular calcium + trypsin?) to prepare the cell for FACS?

It is not clear why the CDH labelling was not detected by FACS. As the reviewer suggests, there could be several reasons: E-cadherin could be broken down by the dissociation reagent (Accutase), or recycled into the cell following the loss of adhesion and the low extracellular calcium in PBS. However, the C-terminal intracellular tail of E-cadherin was tagged, which should not be affected by Accutase. Moreover, recycling into the cell should still result in a detectable fluorescent signal. Notably, the flow cytometry experiments were done as quickly as possible after dissociation to minimize the time that E-cadherin could be degraded or recycled. We also resuspended the cells in MTeSR Plus media instead of PBS, and compared cells grown on iMatrix511 to those grown on Matrigel in case differences in the extracellular matrix affected Ecadherin expression. Another possibility is that the microscopy used for detection of E-cadherin in cells involved using a sweptfield livescan confocal microscope with high NA objective, 100mW 488nm laser and an EMCCD camera with high sensitivity, and perhaps this combination permitted detection better than the detector on the BD FACSMelody used for FACs.

(4) The authors write that the "Tubulin was cytosolic during interphase" which is surprising (and see also figure 3H), as I was expecting it to be incorporated in microtubules. May this be an issue of insufficient resolution (if I'm right this was imaged with 20x, NA=0.35 and so the resolution could be improved by imaging at higher NA)?

Indeed, as the reviewer points out, our terminology (cytosol vs. microtubule) reflects the low resolution of the imaging for the cell populations in dishes and the individual alpha-tubulin monomers being labelled with the mNG211 tag, which are present as cytoplasmic monomers as well as polymers on microtubules. However, even in this image (Fig. 2C), the mitotic spindle microtubules are visible as they are so robust compared to the interphase microtubules. Notably, when we imaged cells from the cloned tagged cell line using a microscope designed for live imaging with a higher NA objective (see above), endogenous tagged TUBA1B was even more clearly visible in spindle microtubules, and was weakly observed in some microtubules in interphase cells, although they are slightly out of focus (Fig. 3H). If we had focused on a lower focal plane where the interphase cells are located and altered the optical settings, we would see more microtubules.

(5) It would be nice to have access to the Timelapse data as supplemental movies (.e.g from the experiments shown in Figure 4).

We have added the movies corresponding to the timeplase images as supplementary movies (Movies S1-6), with the raw and restored movies shown side-by-side.

(6) In Figure 3B, the order of the colors in the bar is reversed relative to the order of the legend. Would it be possible to use the same order? That makes it easier for me (as a colorblind person) to match the colors in the figure with that of the legend.

We have modified the legend in Fig 2B and 3B to be in the same order as the bars.